JCB Journal of Cell Biology

## REPORT

# Borealin–nucleosome interaction secures chromosome association of the chromosomal passenger complex

Maria A. Abad[1], Jan G. Ruppert[1]*, Lana Buzuk[1]*, Martin Wear[1], Juan Zou[1], Kim M. Webb[1], David A. Kelly[1], Philipp Voigt[1], Juri Rappsilber[1,2], William C. Earnshaw[1], and A. Arockia Jeyaprakash[1]

Chromosome association of the chromosomal passenger complex (CPC; consisting of Borealin, Survivin, INCENP, and the Aurora B kinase) is essential to achieve error-free chromosome segregation during cell division. Hence, understanding the mechanisms driving the chromosome association of the CPC is of paramount importance. Here using a multifaceted approach, we show that the CPC binds nucleosomes through a multivalent interaction predominantly involving Borealin. Strikingly, Survivin, previously suggested to target the CPC to centromeres, failed to bind nucleosomes on its own and requires Borealin and INCENP for its binding. Disrupting Borealin–nucleosome interactions excluded the CPC from chromosomes and caused chromosome congression defects. We also show that Borealin-mediated chromosome association of the CPC is critical for Haspin- and Bub1-mediated centromere enrichment of the CPC and works upstream of the latter. Our work thus establishes Borealin as a master regulator determining the chromosome association and function of the CPC.

## Introduction

Chromosome segregation is a complex process involving numerous protein–protein and protein–DNA interactions tightly controlled by signaling networks consisting of kinases and phosphatases (Funabiki and Wynne, 2013; Gelens et al., 2018; Saurin, 2018). Aurora B kinase, the enzymatic core of the chromosomal passenger complex (CPC), is a key regulator essential for error-free chromosome segregation and functions by controlling multiple steps of cell division: chromosome condensation and cohesion, kinetochore-microtubule attachments, the spindle assembly checkpoint, and cytokinesis (Carmena et al., 2012; Funabiki and Wynne, 2013; Hindriksen et al., 2017). The CPC is composed of Aurora B, inner centromere protein (INCENP), Borealin/Dasra, and Survivin and can be divided into distinct localization and kinase modules, linked by the central helical coil of INCENP. The CPC localization module (CPC_LM), consisting of Borealin, Survivin, and the first 58 aa of INCENP, controls the localization of the CPC (Klein et al., 2006; Jeyaprakash et al., 2007). The kinase module consists of Aurora B and the IN-box of INCENP, a well-conserved C-terminal region required for full activation of Aurora B kinase (Bishop and Schumacher, 2002; Honda et al., 2003; Sessa et al., 2005). CPC function is tightly linked to its distinct localization during different stages of cell division. During early stages of mitosis, the CPC localizes to chromosome arms, where it influences chromosome condensation (Lipp et al., 2007; Takemoto et al., 2007). It subsequently concentrates at the inner centromere, where it releases incorrect attachments and regulates the timing of mitotic progression via the spindle assembly checkpoint (Hindriksen et al., 2017). During anaphase, the CPC associates with the central spindle, and during cytokinesis, with the equatorial cortex and midbody to control cell abscission (Cooke et al., 1987; Adams et al., 2000; Uren et al., 2000; Gassmann et al., 2004; Trivedi and Stukenberg, 2016).

CPC localization at centromeres has been suggested to depend on the coexistence of two histone modifications: Haspin-mediated phosphorylation on histone H3 Thr3 (H3T3ph) and Bub1-mediated phosphorylation on histone H2A Thr120 (H2AT120ph). According to the proposed models, H3T3ph is directly recognized by the BIR domain of Survivin (Kelly et al., 2010; Wang et al., 2010; Yamagishi et al., 2010) and H2AT120ph is read by hSgo1, which then recruits Borealin (Tsukahara et al., 2010; Yamagishi et al., 2010). However, Haspin depletion by siRNA does not abolish CPC association to chromosomes (Wang et al., 2010), and Survivin is not sufficient to achieve centromeric enrichment or chromosomal association of CPC in cells expressing Borealin lacking its C-terminal half (Jeyaprakash et al., 2007). These observations collectively highlight a central question that remains unanswered: How does the CPC associate

[1]Wellcome Centre for Cell Biology, University of Edinburgh, Edinburgh, Scotland, UK; [2]Technical University, Berlin, Germany.

*J.G. Ruppert and L. Buzuk contributed equally to this paper; Correspondence to A. Arockia Jeyaprakash: jeyaprakash.arulanandam@ed.ac.uk.

with chromosomes during early prophase before its H3T3ph-mediated centromeric enrichment?

## Results and discussion

### Borealin nucleosome binding is essential for chromosome association of the CPC

Consistent with our previous observations (Jeyaprakash et al., 2007), transient expression of a Myc-tagged Borealin lacking the first 9 aa and the C-terminal half (Myc-Borealin$_{10-109}$) failed to rescue the siRNA-mediated depletion of endogenous Borealin, and the CPC was completely excluded from chromosomes during the early stages of mitosis, leading to chromosome congression defects (Fig. 1, A and B; and Fig. S1, A and B). This led us to hypothesize a direct role for Borealin in mediating CPC–nucleosome interactions. To test this, we reconstituted nucleosome core particles (NCPs) containing homogeneous H3T3ph modification and performed electrophoretic mobility shift assays (EMSAs) with recombinant CPC_LM (Borealin-Survivin-INCENP$_{1-58}$; Fig. S1 C). The CPC_LM showed clear binding to modified NCPs as evidenced by its retarded mobility (Fig. 1 C). Interestingly, CPC_LM containing Borealin$_{10-109}$ failed to interact with NCPs even when mixed at a 32 times molar excess (Fig. 1 C). This is particularly surprising, as several studies have shown previously that Survivin can bind synthetic N-terminal HH3 peptides phosphorylated at Thr3 through its BIR domain (Kelly et al., 2010; Wang et al., 2010; Yamagishi et al., 2010). To test if Survivin on its own could bind H3T3ph tail in the context of NCPs, we analyzed the binding of purified Survivin and H3T3ph NCPs by EMSA. Strikingly, Survivin did not bind H3T3ph NCPs (Fig. 1 D), possibly due to a lack of H3 tail accessibility within the NCP. Together, these data demonstrate Borealin to be a major contributor to CPC–nucleosome interactions. Considering Borealin's direct role, we next asked if the CPC_LM can bind unmodified NCPs. Interestingly, the CPC retarded the mobility of unmodified NCPs in the EMSA assays, confirming its binding to unmodified NCPs (Fig. 1 E). These observations establish that the CPC can bind NCPs in a H3T3ph-independent manner, and the interaction is mainly mediated by Borealin.

As H3T3ph has been proposed to be critical for concentrating the CPC at inner centromeres, we speculated that phosphorylation on H3T3 might positively influence nucleosome binding affinity of the CPC. To address this, we performed surface plasmon resonance (SPR) experiments by flowing CPC at different concentrations over the sensor surface containing immobilized NCP and determined steady-state binding affinities. While the CPC_LM interacted with unmodified NCPs with a dissociation constant ($K_d$) of 295.2 ± 40.9 nM, interaction with H3T3ph NCPs was threefold tighter, at 102.8 ± 34.2 nM (Fig. 1 F). Interestingly, this increase in affinity for phosphorylated NCPs was due to Survivin binding to the H3 N-terminal tail, as a CPC containing a Survivin BIR mutant deficient for binding the phosphorylated H3 tail (CPC_LM$_{SUR MUT}$; Fig. S1 F), bound both modified and unmodified NCPs with a similar affinity (Fig. 1 F). Consistent with this, the NCP reconstituted with H3 lacking the first 31 aa (NCP$_{H3 Tailless}$) bound CPC to form a robust CPC-NCP$_{H3 Tailless}$ complex in SEC and EMSA assays (Fig. S1, D

and E). Thus, our data show that although the affinity of the CPC for modified NCPs is enhanced, the Survivin–H3 interaction is not essential for CPC nucleosome binding per se. Moreover, transient expression of Survivin BIR mutant (K62/E65/H80A; GFP-Survivin$_{MUT}$) not capable of binding the phosphorylated Histone H3 tail did not abolish the chromosome association of the CPC in the Survivin siRNA rescue assay (Fig. S1, G and H). This agrees with the previous studies (Cao et al., 2006; Niedzialkowska et al., 2012) where disrupting survivin–H3 interaction reduced centromere association of the CPC; however, it did not abolish the chromosome association of the complex. Overall, we conclude that Borealin-mediated CPC–nucleosome interaction is essential for the chromosome association of the CPC.

### N-terminal 9 aa and C-terminal half of Borealin are required for CPC–nucleosome interaction

Considering the essential contribution of Borealin towards nucleosome binding, we next mapped the regions of Borealin directly involved in nucleosome interaction. We reconstituted several versions of CPC_LM complexes containing different Borealin mutants (designed based on its domain architecture; Fig. 1 A and Fig. S2, A and B) and tested them in EMSA assays with and without H3T3 phosphorylation on NCPs (Fig. 2 A). Deleting a well-conserved unstructured central region of Borealin (amino acid residues 110–206, CPC_LM$_{BOR Δloop}$; Fig. S2 C) abolished CPC binding to NCP almost completely. The deletion of either the N-terminal 9 aa (CPC_LM$_{BOR 10-end}$) or that in combination with the C-terminal 59 aa of Borealin (CPC_LM$_{BOR 10-221}$) also caused a noticeable reduction in binding (Fig. 2 A). Deleting just the C-terminal 59 aa of Borealin (CPC_LM$_{BOR 1-221}$) appeared to bind NCPs with a slightly increased efficiency compared with the CPC_LM. We speculate that removing just the C-terminal region of Borealin perhaps modulates the accessible CPC surface, facilitating additional nonspecific interactions with NCPs or free DNA in the EMSAs, where the low-ionic-strength conditions can strengthen ionic/salt bridge contacts. Considering the qualitative nature of the EMSA assay, we evaluated the NCP-binding affinities of mutant CPC complexes in SPR assays (Figs. 2 B and S2 D). CPC_LM$_{BOR 10-end}$ showed threefold reduction in binding affinity compared with CPC_LM ($K_d$ = 860 ± 89.5 vs. 295.2 ± 40.9 nM). Both CPC_LM$_{BOR Δloop}$ and CPC_LM$_{BOR 10-221}$ exhibited even weaker NCP binding, with measured affinities in the micromolar range (Fig. 2 B). Together our data show that the N-terminal 9 aa and C-terminal half of Borealin contribute to nucleosome binding, possibly through multiple physical contacts.

Having dissected the contribution of Borealin for nucleosome binding in vitro, we evaluated the behavior in vivo of Borealin mutants showing reduced NCP binding in siRNA rescue experiments (Fig. 2, C–E). Transient expression of Myc-Borealin$_{10-end}$ in Borealin siRNA-depleted HeLa cells resulted in a 50% reduction of the centromeric association of the CPC. In contrast, expression of Myc-Borealin$_{Δloop}$ or Borealin$_{10-221}$ resulted in almost complete exclusion of the CPC from chromosomes (Fig. 2, C and D), irrespective of the expression levels of

Figure 1. **Borealin nucleosome binding is essential for chromosome association of the CPC. (A)** Domain architecture of the subunits of the CPC. **(B)** Representative fluorescence images of a siRNA rescue assay for Borealin$_{10-109}$ fragment. Immunofluorescent staining of Myc and Survivin in HeLa cells cotransfected with siRNA duplexes targeting the 3' UTR region of Borealin and Myc-Borealin constructs. Hoechst was used for DNA staining. Scale bar, 10 µm. All cells transfected with the siRNA and Myc-Borealin$_{10-109}$ fragment showed exclusion of the CPC complex from the chromatin. **(C and E)** Native PAGE analysis of EMSA assays performed with increasing concentrations of recombinant CPC_LM containing different Borealin fragments and either 20 nM phosphorylated (H3T3ph; C) or unmodified IR700-labeled (E) NCPs. **(D)** EMSA assays performed with increasing concentrations of Survivin with 20 nM IR700-labeled H3T3ph NCPs. **(F)** Representative SPR sensorgrams of the interaction between different CPC_LM complexes (CPC_LM, CPC_LM$_{SUR\ MUT}$) or Survivin and unmodified (top) or H3T3ph (middle) NCPs or DNA (bottom) immobilized on the surface of a neutravidin sensor chip. Mean values ($n \geq 3$, ±SEM) determined for the equilibrium $K_d$ are shown in boxes underneath the sensorgrams. For a detailed description of the CPC domain architecture, refer to Fig. S1 C.

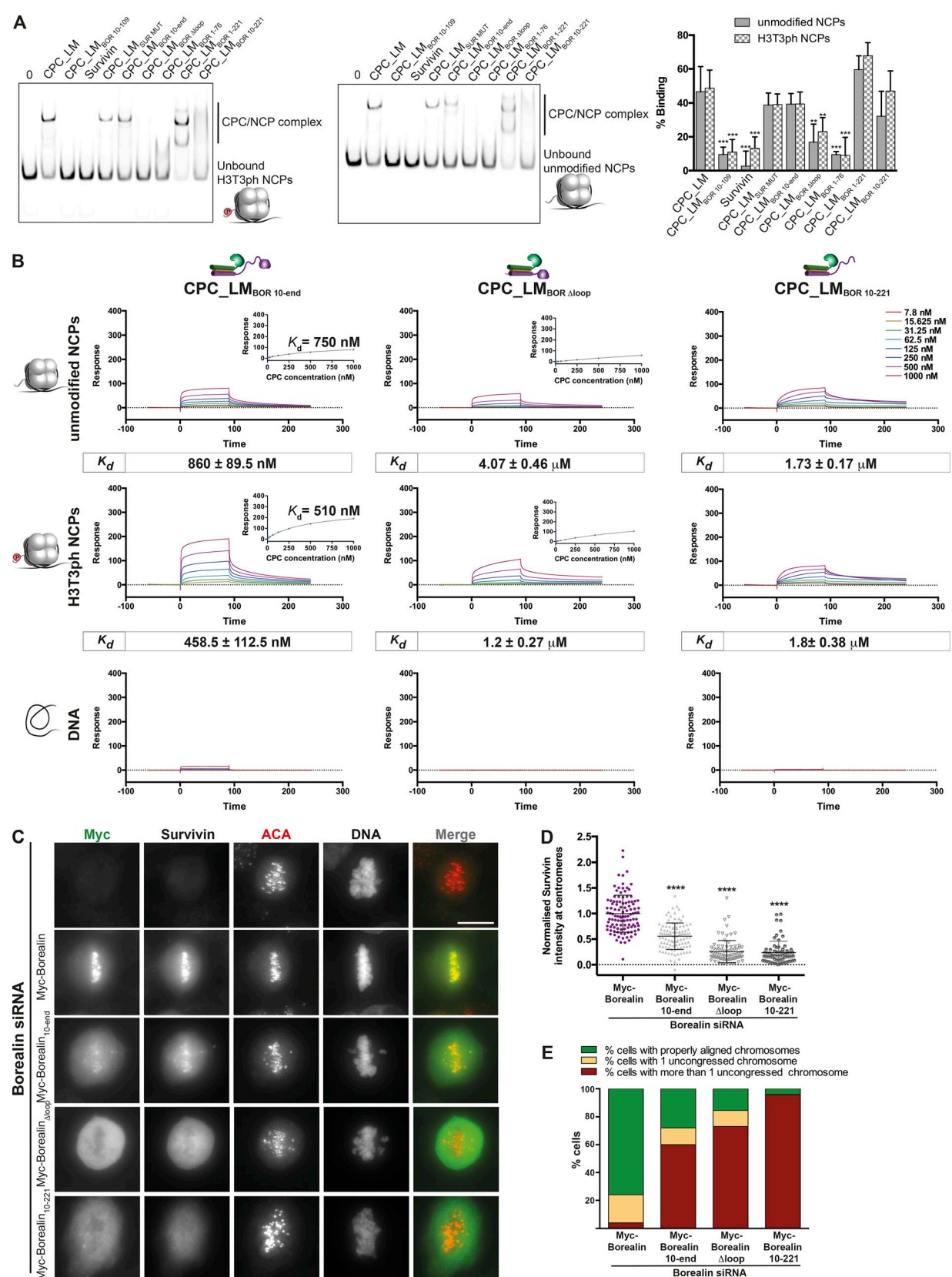

Figure 2. **N-terminal nine amino acids and C-terminal half of Borealin are required for CPC–chromatin interaction. (A)** Native PAGE analysis of EMSA assays performed with recombinant CPC_LM containing different Borealin truncations binding to IR700-labeled H3T3ph NCPs (left) and unmodified NCPs (middle) and quantification of binding (right). Concentrations of the NCP and the CPC used in the assay were 20 and 160 nM, respectively. Mean of percentage of binding ± SD; $n = 5$; **, $P \leq 0.01$; ***, $P \leq 0.001$, unpaired $t$ test. **(B)** Representative SPR sensorgrams of the interaction between different CPC_LM complexes

(CPC_LM$_{BOR\ 10-end}$, CPC_LM$_{BOR\ \Delta loop}$, and CPC_LM$_{BOR\ 10-221}$) and unmodified (top) or H3T3ph (middle) NCPs or DNA (bottom). Mean values ($n \geq 3$, ±SEM) determined for the equilibrium $K_d$ are shown in boxes underneath the sensorgrams. For a detailed description of the CPC domain architecture, refer to Fig. S1 C. **(C)** Representative fluorescence images of a rescue assay for Borealin, Borealin$_{10-end}$, Borealin$_{\Delta loop}$, and Borealin$_{10-221}$ constructs. Immunofluorescent staining of Myc, Survivin, and ACA in HeLa cells cotransfected with siRNA duplexes targeting the 3' UTR region of Borealin and Myc-Borealin constructs. Hoechst was used for DNA staining. Scale bar, 10 μm. **(D)** Quantification of Survivin levels at the centromeres for the siRNA-rescue assays with Myc-Borealin ($n = 111$ cells), Myc-Borealin$_{10-end}$ ($n = 101$ cells), Myc-Borealin$_{\Delta loop}$ ($n = 85$ cells), and Borealin$_{10-221}$ ($n = 72$ cells) shown in C (three independent experiments, mean ± SEM, Mann–Whitney $U$ test; ****, P < 0.0001). **(E)** Quantification of uncongressed chromosomes observed for the siRNA-rescue assay of Myc-Borealin, Myc-Borealin$_{10-end}$, Myc-Borealin$_{\Delta loop}$, and Myc-Borealin$_{10-221}$ fragments shown in C. A minimum of 25 cells were counted for each construct.

the constructs in the individual cells. All Myc-Borealin constructs expressed at similar levels (Fig. S2 E) and interacted with other CPC subunits (Survivin, INCENP, and Aurora B; Fig. S2 F). We analyzed the mitotic defects caused by these Borealin mutants by quantifying chromosome congression defects. All Borealin mutants that showed reduced nucleosome binding resulted in an increase in the number of cells with uncongressed chromosomes (Fig. 2 E). Myc-Borealin$_{10-221}$ showed a noticeably stronger phenotype, with almost all cells showing more than one uncongressed chromosome. This suggests that the C-terminal region of Borealin might have an additional role in ensuring proper chromosome congression. Collectively, these observations demonstrate that Borealin-mediated nucleosome binding is essential for chromosome association of the CPC in vivo.

## CPC–nucleosome binding is mediated by multivalent interactions predominantly involving Borealin

To gain structural insight into the underlying mechanism, the CPC–NCP complexes (Fig. S3 A) were cross-linked using 1-ethyl-3-(3-dimethylaminopropyl) carbodiimide (EDC) and analyzed by mass spectrometry (MS; Fig. S3, B and C). Consistent with our in vitro binding studies, regions of Borealin shown here to be critical for nucleosome binding made extensive contacts with NCPs, whereas Survivin interactions were mostly limited to the BIR domain and the H3 N-terminal tail (Figs. 3 A and S3 D). We note that the intersubunit cross-links observed generally agree with the available 3D structures. Mapping the cross-links onto the available 3D structures of NCP and CPC (Fig. 3 B) suggested interactions (a) involving multiple Borealin regions (positively charged N-terminal region, central loop region, and the C-terminal dimerization domain) and the NCP acidic patch formed between H2A and H2B, a surface commonly involved in nucleosome recognition (Barbera et al., 2006; Roussel et al., 2008; Makde et al., 2010; Armache et al., 2011; Kato et al., 2011); and (b) between Borealin loop residues and histone residues present along the DNA–histone interface. Considering the flexibility associated with the Borealin loop (96 aa long) we speculated that some of the cross-links involving Borealin C-terminal region (the central loop and the dimerization domain) and the NCP acidic patch might represent nonspecific/transient interactions, while the contacts involving the Borealin N-terminus might facilitate docking of the positively charged CPC triple helical bundle at the acidic patch. Such a mode of binding is likely to position the Survivin BIR domain for efficient H3 tail binding.

First, we assessed the contribution of Borealin N-terminus by evaluating the NCP binding of CPC_LM containing Borealin K12/R17/K20E mutant (CPC_LM$_{BOR\ K12/R17/K20E}$; designed based on the cross-linking data shown in Fig. 3 B) in SPR experiments. Borealin K12/R17/K20E mutant reduced NCP-binding of the CPC by approximately twofold compared with the CPC_LM ($K_d = 528 \pm 44.2$ vs. $295.2 \pm 40.9$ nM), and the corresponding value for the H3T3ph NCP is approximately fivefold ($K_d = 672.3 \pm 70.5$ vs. $102.8 \pm 34.2$; Figs. 4 A and S3 E). Consistent with the in vitro binding data, transient expression of Myc-Borealin$_{K12/R17/K20E}$ in Borealin siRNA-depleted cells showed a 50% decrease in the centromeric levels of the CPC (Fig. 4, B and C; and Fig. S3 F). We conclude that Borealin N-terminal region interaction with NCP acidic patch is critical for efficient CPC–NCP interaction in vitro and CPC chromosome association in vivo.

We next aimed to understand the molecular basis for the contribution of the Borealin loop region. As most Borealin loop contacts are near the DNA (the theoretical Isoelectric point [pI] of the loop is 10.4), and as Borealin has previously been proposed to bind DNA (Klein et al., 2006), we hypothesized that the Borealin loop region is directly interacting with nucleosomal DNA. To test this, we revisited the SPR sensorgrams obtained for the DNA binding of CPC_LM and CPC_LM$_{\Delta loop}$ (Figs. 1 F and 2 B). Although CPC_LM clearly interacted with the DNA, the binding affinity was too weak to be accurately determined over the concentration range analyzed. By normalizing the steady-state DNA binding level for 1 μM CPC_LM and comparing the molecular weight–corrected response, we estimated the apparent affinity for DNA binding of CPC_LM and the CPC_LM$_{\Delta loop}$. CPC_LM bound DNA with an apparent $K_d$ of ~5 μM, while the corresponding interaction for the CPC_LM$_{\Delta loop}$ was very much weaker, with a $K_d$ of ~20 μM, indicating that the loop region is responsible for the binding of CPC to the DNA (Fig. 4 D). Furthermore, we also tested the contribution of different Borealin loop regions to bind DNA in the SPR assays using recombinant Sumo-Borealin$_{110-206}$ (spanning the entire loop), Sumo-Borealin$_{110-188}$ (N-terminal half of the loop), and Sumo-Borealin$_{189-206}$ (C-terminal half of the loop). Supporting our hypothesis, Sumo-Borealin$_{110-206}$ bound DNA with an apparent $K_d$ of ~5 μM. Removing the N-terminal half of the loop, Sumo-Borealin$_{189-206}$, bound DNA with a strongly reduced affinity ($K_d$ ~20 μM), while removing the C-terminal half of the loop, Sumo-Borealin$_{110-188}$, showed only a modest reduction in affinity ($K_d$ ~10 μM; Figs. 4 E and S3 G). Consistent with this, Borealin mutants either lacking the C-terminal half of the loop (CPC_LM$_{\Delta189-206}$) or harboring point mutations in this region (Borealin K198/T199E; CPC_LM$_{BOR\ K198/T199E}$) did not show a noticeable reduction in the nucleosome binding ability of CPC

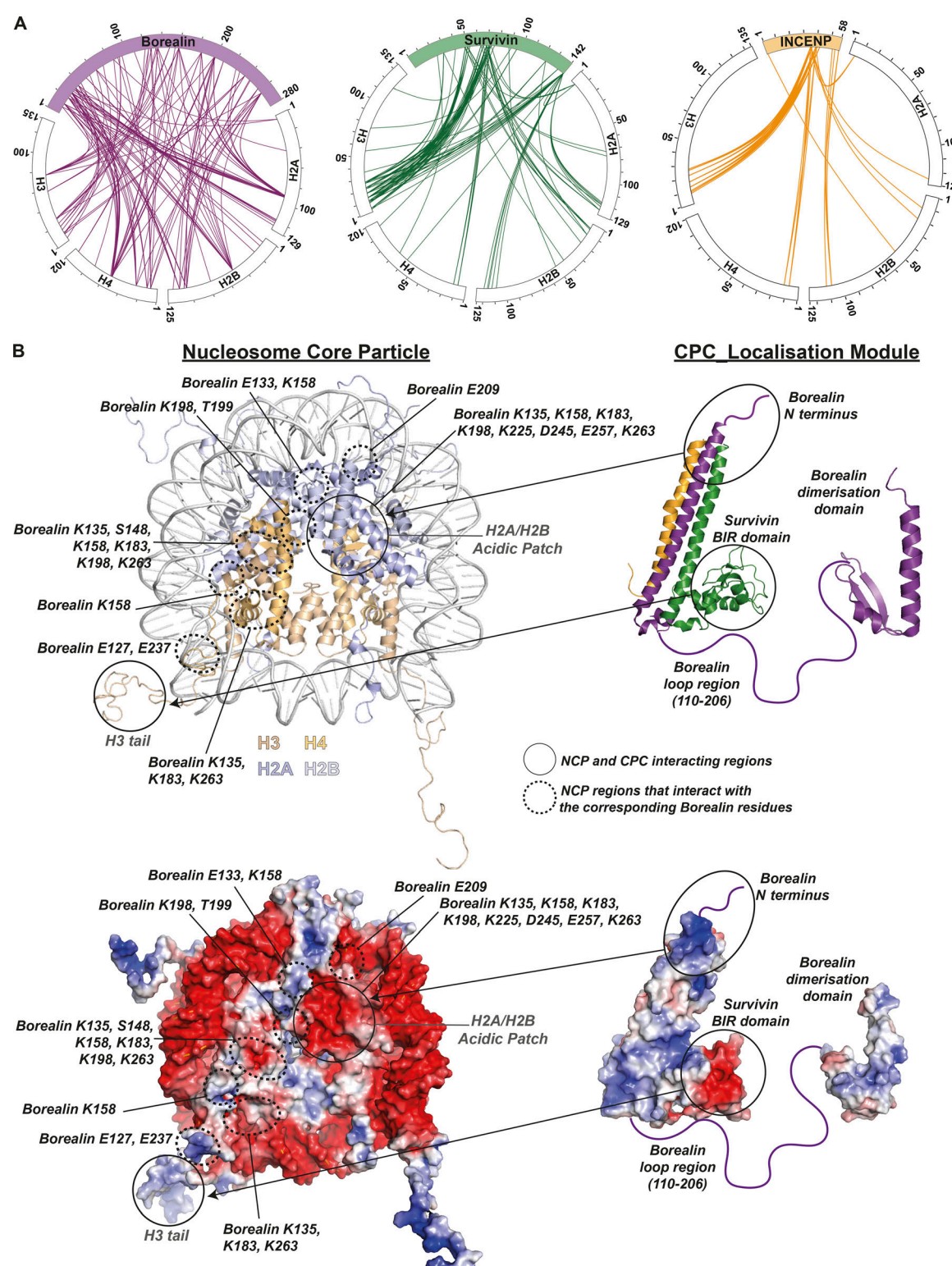

Figure 3. **CPC–nucleosome binding is mediated by multivalent interactions predominantly involving Borealin. (A)** Circle view of the cross-links observed between the subunits of the CPC (Borealin, purple; Survivin, green; INCENP, yellow) and histones from unmodified NCPs. Intermolecular contacts involving Borealin, Survivin, and INCENP and histones are shown as purple, green, or yellow lines, respectively, using XiNET (Kolbowski et al., 2018). **(B)** Cartoon representation of the crystal/NMR structures of the NCP (PDB: 1KX5; Davey et al., 2002) and CPC (CPC core PDB: 2QFA [Jeyaprakash et al., 2007] and Borealin dimerization domain PDB: 2KDD [Bourhis et al., 2009]; top). Surface representation of the NCP and the CPC colored based on the electrostatic surface potential calculated using APBS in Pymol v2.0.6 (bottom).

Figure 4. **Borealin N-terminal region and the central loop contribute to NCP binding by interacting with the NCP acidic patch and DNA. (A)** Representative SPR sensorgrams of the interaction between the cross-linking–based CPC_LM mutants (CPC_LM$_{BOR\ K12/R17/K20E}$, CPC_LM$_{BOR\ K135/K158/K183/K198/T199E}$, and CPC_LM$_{BOR\ K198/T199E}$) and unmodified (top) or H3T3ph (middle) NCPs or DNA (bottom). Mean values ($n \geq 3$, ±SEM) determined for the equilibrium $K_d$ are

shown in boxes underneath the sensorgrams. **(B)** Representative fluorescence images of rescue assays for Myc-Borealin, Myc-Borealin K12/K17/K20E, and Myc-Borealin K135/K158/K183/K198/T199E constructs in HeLa cells depleted of Borealin by siRNA. Immunofluorescent staining of Myc, Survivin, and ACA. Hoechst was used for DNA staining. Scale bar, 10 µm. **(C)** Quantification of Survivin levels at the centromeres for the siRNA-rescue assays with Myc-Borealin ($n$ = 41 cells), Myc-Borealin K12/R17/K20E ($n$ = 42 cells), and Myc-Borealin K135/K158/K183/K198/T199E ($n$ = 43 cells) shown in B (three independent experiments, mean ± SEM, Mann–Whitney $U$ test; ****, P < 0.0001). **(D)** Quantification of DNA binding by CPC_LM and CPC_LM$_{\Delta loop}$. Binding was normalized to the steady-state response for 1 µM CPC_LM ($n$ = 4). **(E)** Quantification of DNA binding by Sumo-Borealin$_{110-206}$, Sumo-Borealin$_{110-188}$, and Sumo-Borealin$_{189-206}$ ($n \geq$ 2). Binding was normalized to the steady-state response for 1 µM Sumo-Borealin$_{110-206}$. Representative sensorgrams are shown in inset. The apparent $K_d$ values for each sample are indicated and were estimated from molecular weight–corrected responses in comparison to CPC_LM binding to unmodified NCPs. **(F)** Representative SPR sensorgrams for the interaction between Sumo-Borealin$_{222-280}$ and NCPs ($n$ = 1).

($K_d$ = 260 ± 37 and 400 ± 30.3 nM, respectively; Figs. 4 A and S3 H). However, either when these mutations were combined with additional mutations within the N-terminal half of the loop (K135/K158/K183/K198/T199E; CPC_LM$_{BOR~K135/K158/K183/K198/T199E}$) or when the N-terminal half of the loop was deleted (CPC_LM$_{\Delta 110-188}$) nucleosome binding ability of the CPC decreased strongly ($K_d$ = 1.1 ± 0.2 and 1.8 ± 0.2 µM, respectively; Figs. 4 A and S3 H). In agreement with the in vitro data, Myc-Borealin K135/K158/K183/K198/T199E showed a strong reduction in the chromosome association of the CPC (Fig. 4, B and C). We conclude that the Borealin loop can directly bind DNA and nearby histone residues along the DNA–histone interface with interactions mainly involving the N-terminal half of the loop region (amino acid residues 110–188).

Finally, to understand the contribution of the Borealin dimerization domain, we tested NCP binding ability of the Borealin dimerization domain (Sumo-Borealin$_{222-280}$) by SPR, which did not show any binding (Fig. 4 F). However, the steady-state Maximum Response Unit (RUmax) values observed experimentally for CPC_LM binding to NCP (∼275 RU; Fig. 1 F) were very close to the theoretical RUmax values expected for a 1:1 stoichiometric interaction (∼250–300 RU) between a CPC_LM dimer (100 kD) and an NCP (200 kD), given the molecular weight ratio of the interacting components and the amount of nucleosome immobilized on the sensor surface (500 RU). Based on this, we speculate that Borealin dimerization domain likely increases the affinity by modulating the interaction by providing two copies of CPC that can interact with the acidic patch and H3 tail present on either side of a single NCP.

Taken together, the data presented here suggest a model where the CPC_LM with the highly basic Borealin N-terminal region docks onto the NCP acidic patch. This interaction may orient the Survivin BIR domain to facilitate binding of H3T3ph. The Borealin loop likely binds along the DNA–histone interface by directly interacting with DNA and nearby histone residues, while Borealin dimerization enhances nucleosome affinity by facilitating full occupation of the CPC-interaction sites (acidic patch, DNA, and H3 tail) related by the intrinsic twofold symmetry of the NCP.

### Borealin-mediated chromosome association of the CPC is an upstream requirement for its Haspin- and Bub1-mediated centromeric enrichment

As Haspin activity has been suggested to be stimulated by the Aurora B kinase, we next evaluated the impact of Borealin-mediated chromosome association of CPC on Haspin activity. Strikingly, in Borealin-depleted cells, H3T3 phosphorylation was reduced to low levels (Fig. 5 A). While the expression of Myc-Borealin rescued these H3T3ph levels, expression of a Borealin mutant incapable of chromosome association (Myc-Borealin$_{\Delta loop}$) failed to do so. Likewise, Borealin depletion led to a decrease in the levels of H2AT120 phosphorylation (Fig. 5 B), which could be rescued with Borealin, but only partially with Borealin$_{\Delta loop}$. Notably, Haspin depletion, which resulted in reduced H3T3 phosphorylation and as a consequence diffused CPC localization along the chromosome arms, did not affect H2AT120 phosphorylation (Fig. S3 I). Moreover, Bub1 siRNA, which led to a reduction of H2AT120 phosphorylation, did not affect H3T3 levels, confirming that H3T3ph is upstream of H2AT120 phosphorylation (Fig. S3 I). Together, our data demonstrate that CPC binding to nucleosomes is an upstream requirement for Haspin and Bub1 activities and for Haspin/Bub1-mediated CPC enrichment at centromeres.

In summary, our data suggest a mechanism for the chromosome association of the CPC essential for error-free chromosome segregation (Fig. 5 C). During early mitosis, when there is little or no H3T3ph (Ruppert et al., 2018), the CPC binds chromosomes in a histone modification–independent manner, mainly via Borealin interactions involving multiple contacts with the histone octamer and DNA. Chromatin association of the CPC activates Haspin and Bub1 through Aurora B–mediated Haspin phosphorylation and Bub1 recruitment (Krenn and Musacchio, 2015; Hindriksen et al., 2017), which in turn phosphorylates H3T3 and H2AT120, respectively. These histone phosphorylation marks increase the affinity of CPC for chromosomes due to Survivin BIR interaction with H3T3ph (Kelly et al., 2010; Wang et al., 2010; Yamagishi et al., 2010) and possibly binding of Borealin to Sgo1 (Tsukahara et al., 2010; Yamagishi et al., 2010). This facilitates CPC enrichment at inner centromeres during prometaphase and metaphase. In the future, it will be important to understand how the H2AT120ph-hSgo1-Borealin pathway coexists with the Borealin-mediated chromosome association we report here and how multivalent interactions between CPC and chromosomes are weakened to transfer the CPC from chromatin to the central spindle during late metaphase/anaphase (Earnshaw and Cooke, 1991; Cai et al., 2018).

## Materials and methods
### Protein expression and purification of the CPC
Survivin was cloned as a 3C-cleavable His-GFP–tagged protein in a pRSET vector (Thermo Fisher Scientific). The different Borealin fragments were cloned as a TEV-cleavable His-tagged

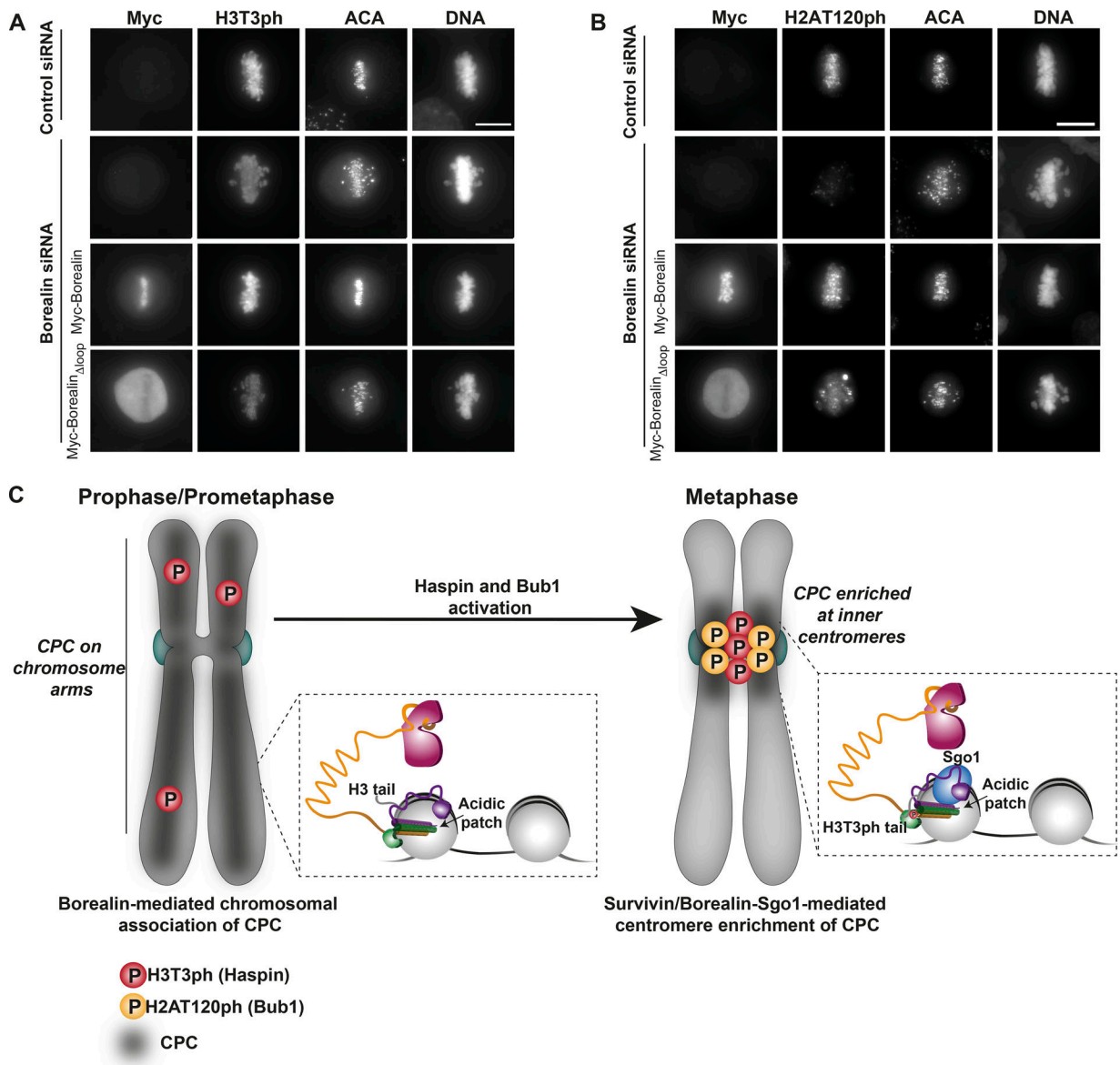

Figure 5. **Borealin-mediated chromosome association of the CPC is an upstream requirement for its Haspin- and Bub1-mediated centromeric enrichment. (A and B)** Immunofluorescence analysis of ACA and H3T3ph (A) or H2AT120ph (B) levels upon Borealin depletion using siRNA duplexes and rescue with different Borealin constructs. Hoechst was used for DNA staining. Scale bar, 10 µm. **(C)** Model for Borealin-mediated chromatin association of the CPC and subsequent centromere enrichment. Considering that Borealin can dimerize and is required for efficient NCP binding, we speculate that two copies of CPC likely bind symmetrically equivalent CPC-interaction sites on NCP. For clarity, just one CPC binding is shown in the cartoon.

protein in a pETM vector (gift from C. Romier, Institute of Genetics and Molecular and Cellular Biology, Strasbourg, France), and INCENP$_{1-58}$ was cloned as an untagged protein in a pMCNcs vector. The Borealin deletions and mutations were generated using the Quikchange site-directed mutagenesis method (Stratagene). Sumo-tagged Borealin loop regions (Sumo-Borealin$_{110-206}$, Sumo-Borealin$_{110-188}$, and Sumo-Borealin$_{189-206}$) and dimerization domain (Sumo-Borealin$_{222-280}$) were cloned into the pET His6 Sumo TEV expression vector with BioBrick polypromoter restriction sites (14-S; a gift from Scott Gradia, California Institute for Quantitative Biosciences [QB3], University of California, Berkeley, Berkeley, CA; Addgene plasmid 48313).

The CPC subunits were coexpressed in BL21(DE) pLysS strain by cotransforming the three vectors containing the individual CPC components. Cultures were grown at 37°C until the OD reached 0.8 and induced overnight at 18°C. Cells were lysed in lysis buffer (25 mM Hepes, pH 7.5, 500 mM NaCl, 35 mM Imidazole, and 2 mM β-mercaptoethanol) and purified using a 5-ml HisTrap HP column (GE Healthcare). The protein-bound column was washed with lysis buffer, followed by wash buffer (25 mM Hepes, pH 7.5, 1 M NaCl, 50 mM KCl, 10 mM MgCl, 35 mM imidazole, 2 mM ATP, and 2 mM β-mercaptoethanol). Elution buffer (25 mM Hepes, pH 7.5, 500 mM NaCl, 500 mM imidazole, and 2 mM β-mercaptoethanol) was used to elute the proteins. Tags were cleaved overnight with 3C and TEV proteases while

dialyzing against 25 mM Hepes, pH 7.5, 150 mM NaCl, and 4 mM DTT at 4°C. The complexes were further purified by cation exchange chromatography (HiTrap SP, GE Healthcare) followed by gel filtration using a Superdex 200 increase 10/300 column (GE Healthcare) preequilibrated with 25 mM Hepes, pH 8, 250 mM NaCl, 4 mM DTT, and 5% glycerol.

Sumo-tagged loop regions and dimerization domain were transformed in BL21(DE) pLysS strain and grown as described above. Cells were lysed in lysis buffer (25 mM Hepes, pH 7.5, 500 mM NaCl, 35 mM imidazole, and 2 mM β-mercaptoethanol) and purified using 5 ml of HisPur Ni-NTA loose resin (Thermo Fisher Scientific). The protein-bound beads were washed with lysis buffer, followed by high-salt buffer (25 mM Hepes, pH 7.5, 1 M NaCl, 35 mM imidazole, and 2 mM β-mercaptoethanol) and eluted using a buffer containing 25 mM Hepes, pH 7.5, 200 mM NaCl, 400 mM imidazole, and 2 mM β-mercaptoethanol. Proteins were further purified by running them in a Superdex 200 increase 10/300 column (GE Healthcare) preequilibrated with 25 mM Hepes, pH 7.5, 200 mM NaCl, 2 mM DTT, and 5% glycerol.

## Expression and purification of recombinant histones and refolding of histone octamers

Human H2A and H2B and *Xenopus laevis* H3 and H4 was purified as described (Luger et al., 1999). Briefly, H2A, H2B, and H3 were expressed in BL21 (DE3) pLysS cells while H4 were expressed in BL21 cells using LB media. The histones were purified from inclusion bodies using a Dounce glass/glass homogenizer. After solubilization of the inclusion bodies, a three-step dialysis against urea dialysis buffer (7 M urea, 100 mM NaCl, 10 mM Tris, pH 8, 1 mM EDTA, and 5 mM β-mercaptoethanol) was performed. The sample was then applied to a HiTrap Q anion exchange column and a HiTrap SP cation exchange column (GE Healthcare). Histones were eluted from the HiTrap SP column using a linear gradient from 100 mM to 1 M NaCl in 7 M urea, 10 mM Tris, pH 8, 1 mM EDTA, and 1 mM DTT. Purified recombinant histones were dialyzed against water containing 5 mM β-mercaptoethanol before lyophilization and storage at −80°C.

To generate histone H3 phosphorylated at threonine 3 by native chemical ligation, histone H3 lacking residues 1–31, containing a threonine-to-cysteine substitution at position 32 and a cysteine-to-alanine substitution at position 110 (H3Δ1–31 MT32C C110A), was expressed and purified as described above. Native chemical ligation reactions with H3Δ1–31 MT32C C110A and the N-terminal H3 peptide ARTPhKQTARKSTGGKAPRKQLAT-KAARKSAPA containing a C-terminal benzyl thioester (Peptide Protein Research) were performed in 6 M guanidine HCl, 250 mM sodium phosphate buffer, pH 7.2, 150 mM 4-mercaptophenylacetic acid, and 50 mM tris(2-carboxyethyl) phosphine for 72 h at room temperature with constant agitation. Reactions were dialyzed three times against 7 M urea, 100 mM NaCl, 10 mM Tris, pH 8, 1 mM EDTA, and 1 mM DTT. Ligated full-length H3T3Ph histone was separated from unligated truncated histone through cation exchange chromatography on a monoS column (GE Healthcare) and then dialyzed against water containing 5 mM β-mercaptoethanol before lyophilization and

storage at −80°C. To generate tailless H3, we used the H3Δ1-31 MT32C C110A histone H3 to generate the octamers.

Histone octamers were obtained as previously described (Luger et al., 1999). Briefly, lyophilized histones were resuspended in unfolding buffer (7 M guanidine HCl, 20 mM Tris, pH 7.5, and 10 mM DTT) and mixed to equimolar ratios. The histone mix was then dialyzed three times against 500 ml of refolding buffer (10 mM Tris, pH 8, 2 M NaCl, 1 mM EDTA, and 5 mM β-mercaptoethanol). The octamers were obtained by running the histone mix on a size-exclusion chromatography column (Superdex 200 increase 10/300, GE Healthcare) preequilibrated with refolding buffer and stored at −80°C.

## NCP reconstitution

A pBS-601 Widom vector was used to amplify the 147-bp 601 Widom positioning sequence with unlabeled, 5′ IR700- or biotin-labeled primers (forward, 5′-ACAGGATGTATATATGTG ACACG-3′, and reverse, 5′-CTGGAGAATCCCGGTGCC-3′). Mononucleosomes were obtained by using the salt gradient dialysis method (Luger et al., 1999). Purified octamers were incubated with the Widom DNA in a 2:1 ratio (octamer:DNA) in 200 ml of refolding buffer using Slide-A-Lyzer MINI dialysis devices (Thermo Fisher Scientific). The ionic strength was then decreased overnight by pumping TE buffer (10 mM Tris, pH 8, 50 mM NaCl, and 1 mM EDTA) into the beaker containing the refolding buffer using a peristaltic pump, followed by a 2-h dialysis into TE buffer. Fully reconstituted mononucleosomes were analyzed using a 6% acrylamide native gel in 0.5× Tris-borate-EDTA (TBE) buffer run at 100 V for 2 h at 4°C. The octamer:DNA ratio was optimized by using different ratios of octamers and DNA and analyzing the reconstitution levels using 6% acrylamide native gels.

## EMSA

Different concentrations of recombinant CPC were added to 20 nM IR700-labeled NCPs in reaction buffer (10 mM Tris, pH 7.5, 100 mM NaCl, 1 mM MgCl$_2$, 1 mM DTT, 1% glycerol, and 0.1 mg/ml BSA). Reactions were incubated 1 h at 4°C and run in a 6% polyacrylamide native gel in 0.5× TBE buffer at 100 V for 2 h at 4°C. The fluorescent bound and unbound NCPs were detected with Odyssey CLx Infrared Imaging System (LI-COR Biosciences). The fluorescent signal of the band corresponding to unbound NCPs was quantified using ImageJ (National Institutes of Health). Values were plotted and statistically analyzed using Prism 6.0 (GraphPad Software). P values were obtained by a Student's *t* test.

## SPR

SPR measurements were performed using a BIAcore T200 instrument (GE Healthcare). Streptavidin sensor chips were purchased from GE Healthcare. Sensor surfaces were primed before ligand immobilization by three sequential 30-s injections of 1 M NaCl and 50 mM NaOH, at 30 μl·min$^{-1}$, followed by extensive washing with running buffer (25 mM Hepes, 250 mM NaCl, 1 mM DTT, and 0.05% Tween-20, pH 8). Biotinylated ligands (in vitro reconstituted NCPs and DNA) in running buffer were immobilized on appropriate flow cells to ~500 RU, by varying

the contact time of a 20-nM solution, with a flow rate of 5 µl·min⁻¹. Immediately before each SPR experiment, CPC complexes were dialyzed against running buffer for 1 h at 4°C. Titration experiments with the indicated analytes were performed at 8°C using a twofold dilution series from 7.8 to 1,000 nM, in running buffer, injected over the sensor surface, at 100 µl·min⁻¹ with 90-s contact and 150-s dissociation times. The sensor surface was regenerated between individual analyte experiments by dissociating any residual formed complex by injecting running buffer for 300 s at 100 µl·min⁻¹. A streptavidin surface without ligand served as reference flow cell for the bulk correction. Due to the complex nature of the interaction and the clear multiphasic nature of the interaction between the CPC constructs and the nucleosomes observed in some cases, kinetic models were not used in the fitting process. Almost all the interactions studied were well fitted by a simple steady-state interaction model. Equilibrium $K_d$ was calculated from the sensorgrams by global fitting of a steady-state, 1:1 interaction model, with mass transport considerations, using analysis software (v2.02) provided with the Biacore T200 instrument. Data were replotted for clarity using Prism 6.0.

## Rescue experiments and immunofluorescence microscopy

The pCDNA3.1 vector containing N-terminally tagged 3xMyc-Borealin was a gift from E. Nigg's laboratory (Biozentrum, The Centre for Molecular Life Sciences, Basel, Switzerland). Truncations and mutations of Borealin were obtained using the Quikchange site-directed mutagenesis method (Stratagene). Depletion of endogenous Borealin and Survivin using RNAi and rescue experiments were performed as previously described (Klein et al., 2006) using jetPRIME (Polyplus Transfection). The oligonucleotides targeting the 3′ UTR of Borealin (5′-AGGTA-GAGCTGTCTGTTCAdTdT-3′) and Survivin (Survivin HP validated siRNA 1027400, SI02652958; Qiagen; Klein et al., 2006) or targeting luciferase as a control (5′-CGUACGCGGAAUA-CUUCGAdTdT-3′; Elbashir et al., 2001) were described previously. RNAi depletion of Haspin was performed using oligonucleotides described previously (Dai et al., 2005; siRNA ID 1093). The oligonucleotide used for the RNAi depletion of Bub1 was 5′-AAGCTTGTGATAAAGAGTCAAdTdT-3′. Cells were transfected using jetPRIME and fixed in 4% PFA 48 h after transfection. All siRNA oligonucleotides were purchased from Qiagen. 50 nM siRNA oligonucleotides and 300 ng of Myc-Borealin vectors were used for transfections. HeLa cells were grown on coverslips in 12-well plates, and medium was changed 12 h after transfection. Cells were fixed in 4% PFA 36 h after transfection.

For quantification of the Survivin signal, HeLa Kyoto or HeLa CDK1 analogue sensitive cells (CDK1-as; Ruppert et al., 2018) were used. CDK1-as cells were synchronized for 14 h using 10 µM 1NM-PP1, and HeLa Kyoto were synchronized using 16-h treatment with 10 µM RO3306 and fixed 90 min after washout to increase the number of cells in metaphase. For the analysis of cells with uncongressed chromosomes, a minimum of 20 mitotic cells were counted.

The following antibodies were used for indirect immunofluorescence: anti-myc (1:200; 9E10; Merck Millipore), anti-Borealin (1:500; 147-3; MBL), anti-Survivin (1:500; NB500-201; Novus), anti-H3T3ph (1:500; 07-424; Upstate), anti-H2AT120ph (1:500; 61195; Active Motif), and anti-ACA (1:300; 15-235; Antibodies Inc.). The secondary antibodies used were FITC-conjugated AffiniPure donkey anti-mouse IgG, TRITC-conjugated AffiniPure goat anti-rabbit, Cy5-conjugated AffiniPure donkey anti-human, FITC-conjugated AffiniPure donkey anti-rabbit, and TRITC-conjugated AffiniPure donkey anti-mouse (1:300; Jackson Immunoresearch). Hoechst 33342 was used for DNA staining. Imaging was performed at room temperature using a wide-field DeltaVision Elite (Applied Precision) microscope with Photometrics Cool Snap HP camera and 100× NA 1.4 Plan Apochromat objective with oil immersion (refractive index = 1.514) using the SoftWoRx 3.6 (Applied Precision) software. Shown images are maximum-intensity projections.

For quantification of the centromeric levels of Survivin, the acquired images were processed by constrained iterative deconvolution using SoftWoRx 3.6 software package (Applied Precision), and the centromere intensity of Survivin was quantified using an ImageJ plugin (https://doi.org/10.5281/zenodo.2574963). Briefly, the plugin quantifies the mean fluorescence signal of Survivin in a 2-pixel-wide ring immediately outside the centromere, defined with the ACA staining. For background subtraction, a selected area within the cytoplasm signal was selected. To compare data from different replicates, values obtained after background correction were averaged and normalized to the mean of Survivin intensity in the Myc-Borealin rescue condition. Statistical significance of the difference between normalized intensities at the centromere region was established by a Mann–Whitney $U$ test using Prism 6.0.

## Immunoprecipitation

HeLa Kyoto cells were grown in 10-cm dishes, transfected, and synchronized using 10 µM RO3306 as described above. 90 min after RO3306 washout, mitotic cells were obtained by shake-off and washed once with PBS, and cell pellets were lysed in lysis buffer (50 mM Tris, pH 7.4, 400 mM NaCl, 40 mM β-glycerol phosphate, 10 mM NaF, 0.5% IGEPAL, 0.1% deoxycholate, 100 µM ATP, 100 µM MgCl₂, 100 nM okadaic acid, 0.3 mM Na-vanadate, and protease inhibitor cocktail tablet [Roche Diagnostics]; adapted from Klein et al., 2006) for 30 min at 4°C with rotation. Lysates were then sonicated and spun down at 15,000 $g$ for 15 min. Beads were prepared by incubating 5 µg of mouse anti-myc antibody (CSB-MA000041Mom; Cusabio) with 30 µl of protein G mag sepharose beads (GE Healthcare) for 10 min at RT. Myc-Borealin proteins were bound to the antibody-coupled beads for 1 h at 4°C with rotation. Beads were then washed three times with 1 ml of lysis buffer, and bound proteins were eluted in 40 µl of sample buffer and analyzed by Western blotting. The antibodies used for the immunoblot were mouse anti-myc (CSB-MA000041Mom; Cusabio), rabbit anti-Survivin (ab469, Abcam), rabbit anti-INCENP (ab12183; Abcam), and mouse anti-Aurora B (611082; BD Transduction Laboratories). All the primary antibodies were used at a 1:1,000 dilution. Secondary antibodies used were goat anti-mouse 680 and donkey anti-rabbit 800 (LI-COR) and were used at 1:2,000 dilution.

Immunoblots were imaged using the Odyssey CLx system (LI-COR).

## Western blot

To study the expression levels of each of the Myc-Borealin constructs, HeLa Kyoto cells were transfected in 12-well dishes as described above and solubilized after 36 h in 1× Laemmli buffer, boiled for 5 min, and analyzed by SDS-PAGE followed by Western blotting. The antibodies used for the immunoblot were rabbit anti-tubulin (1:10,000; ab18251; Abcam), mouse anti-myc (1:1,000; CSB-MA000041Mom, Cusabio), and mouse anti-Borealin (1:1,000; M147-3; MB). Secondary antibodies used were goat anti-mouse 680 and donkey anti-rabbit 800 (LI-COR) at 1:2,000 dilution. Immunoblots were imaged using the Odyssey CLx system.

## Chemical cross-linking and MS analysis

Cross-linking experiments of the CPC-NCP complexes were performed using EDC (Thermo Fisher Scientific) in the presence of N-hydroxysulfosuccinimide (Thermo Fisher Scientific). EDC is a zero-length chemical cross-linker capable of covalently linking primary amines of lysine and the protein N-terminus and to a lesser extend also hydroxyl groups of serine, threonine, and tyrosine with carboxyl groups of aspartate/glutamate. 8 µg of CPC_LM-NCP complexes (0.8 mg/ml) was cross-linked in cross-linking buffer (20 mM Hepes, 100 mM NaCl, 1 mM EDTA, and 2 mM DTT, pH 8) using 30 µg EDC and 66 µg of N-hydroxysulfosuccinimide. The 20-µl reactions were incubated for 2 h at room temperature. The cross-linking was stopped by the addition of 100 mM Tris-HCl. Cross-linking products were resolved using 4–12% Bis-Tris NuPAGE (Invitrogen) for 5 min and briefly stained using Instant Blue (Expedeon). Bands were excised, and the proteins were reduced with 10 mM DTT for 30 min at room temperature, alkylated with 55 mM iodoacetamide for 20 min at room temperature, and digested using 13 ng/µl trypsin (Promega) overnight at 37°C.

The digested peptides were loaded onto C18-Stage-tips (Rappsilber et al., 2007) for liquid chromatography/tandem MS analysis, which was performed using Orbitrap Fusion Lumos (Thermo Fisher Scientific) with a "high/high" acquisition strategy. The peptide separation was performed on an EASY-Spray column (50 cm × 75 µm internal diameter, PepMap C18, 2-µm particles, 100-Å pore size; Thermo Fisher Scientific). Mobile phase A consisted of water and 0.1% vol/vol formic acid. Mobile phase B consisted of 80% vol/vol acetonitrile and 0.1% vol/vol formic acid. Peptides were loaded at a flow rate of 0.3 µl/min and eluted at 0.2 µl/min using a linear gradient going from 2% mobile phase B to 40% mobile phase B over 109 or 139 min (each sample was runs three times with different gradients), followed by a linear increase from 40% to 95% mobile phase B in 11 min. The eluted peptides were directly introduced into the mass spectrometer. MS data were acquired in the data-dependent mode with a 3-s acquisition cycle. Precursor spectra were recorded in the Orbitrap with a resolution of 120,000. The ions with a precursor charge state between 3+ and 8+ were isolated with a window size of 1.6 m/z and fragmented using high-energy collision dissociation with collision energy 30. The fragmentation spectra were recorded in the Orbitrap with a resolution of 15,000. Dynamic exclusion was enabled with single repeat count and 60-s exclusion duration. The MS raw files were processed into peak lists using ProteoWizard (version 3.0.6618; Kessner et al., 2008), and cross-linked peptides were matched to spectra using Xi software (version 1.6.743; Mendes et al., 2018) with in-search assignment of monoisotopic peaks (Lenz et al., 2018). Search parameters were MS accuracy, 3 ppm; MS/MS accuracy, 10 ppm; enzyme, trypsin; cross-linker, EDC; maximum missed cleavages, 4; missing monoisotopic peaks, 2; fixed modification, carbamidomethylation on cysteine; variable modifications, oxidation on methionine and phosphorylation on threonine for phosphorylated sample; fragments, b and y ions with loss of $H_2O$, $NH_3$, and $CH_3SOH$. False discovery rate was computed using XiFDR and results reported at 5% residue level false discovery rate (Fischer and Rappsilber, 2017). The MS proteomics data have been deposited to the ProteomeXchange Consortium via the PRIDE (Perez-Riverol et al., 2019) partner repository with the dataset identifier PXD012882.

## Online supplemental material

Fig. S1 shows the structural organization of the CPC and contains additional data on the contribution of Survivin-H3 tail interaction for CPC nucleosome binding related to Fig. 1. Fig. S2 contains additional information on the recombinant CPC complexes used in the EMSA and SPR assays of Figs. 1 and 2. It also includes the analysis of the expression levels of different myc-tagged Borealin mutants and complex formation with the other CPC subunits by Western blot and immunoprecipitation. Fig. S3 shows additional cross-linking/MS data related to Fig. 3 and Haspin/Bub1 siRNA experiments related to Fig. 5.

## Acknowledgments

We thank the staff of the Edinburgh Protein Production Facility and the Centre Optical Instrumentation Laboratory for their help. We thank E. Nigg for sharing Borealin constructs. We also thank Tony Ly, Dhanya Cheerambathur, and Andrew Goryachev for critical reading of the manuscript.

The Wellcome Trust generously supported this work through a Senior Research Fellowship (202811) to A.A. Jeyaprakash, a Principal Research Fellowship to W.C. Earnshaw (073915), a Senior Research Fellowship to J. Rappsilber (084229), a Sir Henry Dale Fellowship to P. Voigt (104175/Z/14/Z), a Centre Core Grant (092076 and 203149) and an instrument grant (108504) to the Wellcome Trust Centre for Cell Biology, and a Multi-User Equipment grant 101527/Z/13/Z to the Edinburgh Protein Production Facility. J.G. Ruppert was supported by the FP7 People: Marie-Curie Actions PloidyNet, funded by the European Union's Seventh Framework Programme (FP7/2007–2013) under grant agreement number 607722.

The authors declare no competing financial interests.

Author contributions: A.A. Jeyaprakash conceived the project. M.A. Abad, J.G. Ruppert, L. Buzuk, M. Wear, J. Zou, D.A. Kelly, P. Voigt, J. Rappsilber, and W.C. Earnshaw designed the experiments. M.A. Abad, J.G. Ruppert, L. Buzuk, M. Wear, J. Zou, D.A. Kelly, K.M. Webb, and P. Voigt performed the experiments.

M.A. Abad, M. Wear, J.G. Ruppert, W.C. Earnshaw, and A.A. Jeyaprakash wrote the manuscript.

Submitted: 7 May 2019

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
