## [Reviewer comments · The Journal of Cell Biology]

Borealin-Nucleosome Interaction Secures Chromosome Association of the Chromosomal Passenger complex

Maria Alba Abad, Jan Ruppert, Lana Buzuk, Martin Wear, Juan Zou, Kim Webb, David Kelly, Philipp Voigt, Juri Rappsilber, William Earnshaw, and A. Arockia Jeyaprakash

Corresponding Author(s): A. Arockia Jeyaprakash, University of Edinburgh

Review Timeline:

Submission Date:	2019-05-07
Editorial Decision:	2019-05-23
Revision Received:	2019-08-30
Editorial Decision:	2019-09-06
Revision Received:	2019-09-13

Monitoring Editor: Arshad Desai

Scientific Editor: Melina Casadio

Transaction Report:

DOI: <https://doi.org/10.1083/jcb.201905040>

May 24, 2019

Re: JCB manuscript #201905040

Dr. A. Arockia Jeyaprakash
University of Edinburgh
Wellcome Trust Centre for Cell Biology
Max Born Crescent
Edinburgh EH9 3BF
United Kingdom

Dear Dr. Jeyaprakash,

Thank you for submitting your manuscript entitled "Direct Nucleosome Binding of Borealin Secures Chromosome Association and Function of the Chromosomal Passenger Complex". The manuscript was assessed by three expert reviewers, whose comments are appended to this letter. We invite you to submit a revision if you can address the reviewers' key concerns, as outlined here.

You will see that the reviewers find the identification of a role for direct nucleosome binding of Borealin in mediating CPC chromosome localization and function interesting and important. We agree that this new property of Borealin is likely to open up new lines of investigation on the CPC and is a good fit for the JCB Report format. The referees do however have a number of concerns about the data presented that will need to be addressed in a revision. In particular, we recommend that experimental efforts be dedicated to the following:

- 1) Improve the crosslinking & interaction studies along the lines suggested by Revs#1-2 and address inconsistencies between different biochemical methods (EMSA vs. SPR)
- 2) Address Rev#2's point on prior data with a Survivin-INCENP fusion to integrate your new results with the literature
- 3) Address point #1 of Rev#3 which also will aid integrating your findings with prior models
- 4) Reviewer #2's concern articulated in point #1-1. Some effort to improve the cell biological analysis of engineered mutants seems necessary based on reviewer feedback. We believe this is important as the localization analyses link the in vitro biochemical work to the cellular context and are central to the major finding reported in the paper.

We appreciated the reviewers' pertinent questions about your mechanistic model and the downstream consequences (e.g., Rev#1 points #3, 5) but we do not think that addressing the details of Haspin, Bub1 function/recruitment and H3/H2A phosphorylation further are within the scope of this Report. You may consider adding speculation/discussion in the text in response to the reviewer. Please let us know if you anticipate any issues addressing the reviewers' comments or would like to discuss the revision further.

While you are revising your manuscript, please also attend to the following editorial points to help expedite the publication of your manuscript. Please direct any editorial questions to the journal

office.

GENERAL GUIDELINES:

Text limits: Character count for a Report is < 20,000, not including spaces. Count includes title page, abstract, introduction, results, discussion, acknowledgments, and figure legends. Count does not include materials and methods, references, tables, or supplemental legends.

Figures: Reports may have up to 5 main text figures. To avoid delays in production, figures must be prepared according to the policies outlined in our Instructions to Authors, under Data Presentation, <http://jcb.rupress.org/site/misc/ifora.xhtml>. All figures in accepted manuscripts will be screened prior to publication.

IMPORTANT: It is JCB policy that if requested, original data images must be made available. Failure to provide original images upon request will result in unavoidable delays in publication. Please ensure that you have access to all original microscopy and blot data images before submitting your revision.

Supplemental information: There are strict limits on the allowable amount of supplemental data. Reports may have up to 3 supplemental figures. Up to 10 supplemental videos or flash animations are allowed. A summary of all supplemental material should appear at the end of the Materials and methods section.

Our typical timeframe for revisions is three months; if submitted within this timeframe, novelty will not be reassessed at the final decision. Please note that papers are generally considered through only one revision cycle, so any revised manuscript will likely be either accepted or rejected.

Thank you for this interesting contribution to the Journal of Cell Biology. You can contact us at the journal office with any questions, cellbio@rockefeller.edu or call (212) 327-8588.

Sincerely,

Arshad Desai, PhD
Editor, Journal of Cell Biology

Melina Casadio, PhD
Senior Scientific Editor, Journal of Cell Biology

Reviewer #1 (Comments to the Authors (Required)):

Recruitment of the chromosome passenger complex (CPC), comprised of Borealin, Survivin, INCENP and Aurora B, to the inner centromere is required for proper chromosome segregation. The recruitment is thought to depend on a direct interaction of the CPC with the chromatin of the inner

centromere. Understanding the biochemical and structural basis of CPC association with the nucleosome is important, as CPC recruitment to nucleosome is a critical mitotic event. In this short report, Abad et al. use biophysical and cell biological methods to address the mechanism for the chromosome association of CPC. In contrast to what was previously thought, the authors find that CPC binds chromatin in a histone modification independent manner, mainly via Borealin. Binding to H3T3ph is enhanced through Survivin. Based on MS-XL experiments, the authors propose that the interaction of the CPC with the nucleosome involves multiple contacts, especially between Borealin and H3. Based on these data the authors present a potential structural model of CPC recruitment to the nucleosome, although this model is not directly tested. The authors go on to suggest that instead of H3T3ph being upstream of CPC recruitment, that Borealin mediates nucleosome association of the CPC and activates Haspin and Bub1, which in turn phosphorylate H3 Thr3 and H2A Thr120, respectively. Subsequently, these histone phosphorylation marks increase the affinity of CPC for chromatin. The major strength of the manuscript is the strong and well executed biophysical undertaken to answer an important question and that it provides novel evidence for a more refined role for histone H3 modifications in CPC recruitment. However, while the individual observations in the paper are very interesting, the study does not provide a compelling and integrated mechanism for how CPC binding to H3 nucleosome contributes to H3T3ph, and whether this process helps to restrict CPC recruitment.

1. The cross-linking MS is a potentially very interested experiment; however, the manuscript does not provide concrete validation of the observations. The data are consistent with multiple interactions mediating the interaction between the CPC and the nucleosome. And the authors identify potential interaction regions in the NCP and the CPC. However, they do not explain the requirement for the loop region. Moreover, how meaningful the observations are is not clear because there is no in vitro or in vivo experimental validation.
2. The authors compare the in vitro binding of CPC_LM to the modified and unmodified H3 tail but based on the crosslinking MS there are many interactions, and the H3 tail may be inconsequential. H3 tail-less mutants should be tested to determine if H3 tail binding is required.
3. The data in figure 4 suggest that CPC recruitment is required for H3T3ph and H2AT120ph. However, the current manuscript does not provide any mechanistic basis for this pathway. The Borealin loop mutant in figure 4 does not provide significant mechanistic insight into the feed forward mechanism, as it does not get recruited to the inner centromere irrespective of the T3 phosphorylation status.
4. EMSA and SPS results are somewhat inconsistent. BOR (10-end) and (10-221) mutants have no significant effect in nucleosome binding through EMSA, while they appear to be critical in SPR. Granted that the SPR is a much more quantitative measure, but the differences are very stark. In contrast, the 1-221 mutant shows an enhanced binding and is not commented on in the text. The mutant with the largest effect consistent across EMSA and SPR is the delta loop; however, there is not a clear mechanistic reason given for the importance of this domain.
5. It is not clear whether the effects of CPC loss are on Haspin and/or Bub1 recruitment or on the activity of the enzymes. Bub1 and Haspin recruitment should be assayed. Bub1 dependency on H3Th3 phosphorylation should also be tested in the same manner as Haspin is tested for H2ATh120 phosphorylation in figure S5.

Minor issues:

6. The labeling of figure 2A makes it very confusing. It is not immediately clear that the Survivin lane lacks the Borealin and INCENP fragment. In addition, it would be helpful to have a schematic of these mutants.
7. It would be useful to label to the N and C termini in the small diagrams of the CPC above the figure to better orient the reader.

8. Likewise, the insets in the model in figure 4 are very small and extremely difficult to read.
9. The pattern of ACA staining looks different in the Myc-Borealin 10-221 condition in Figure 2C suggesting the phenotype is distinct. The analysis of phenotypes of the different Borealin mutants is limited and could be expanded to include cell cycle, chromosome segregation or mitotic defects that might help distinguish to roles of the domains of Borealin.
10. The delta loop and 10-221 mutants should also be included in statistical part of figure 2C.

Reviewer #2 (Comments to the Authors (Required)):

The Chromosomal Passenger Complex composed of Aurora B, Survivin, Borealin and INCENP plays multiple essential roles during mitosis. One of its key functions is to regulate kinetochore-microtubule attachment and centromeric cohesion through localizing at the centromere. It has been established that this centromeric enrichment of the CPC is mediated by mitosis-specific phosphorylation of two histone residues, H3T3 and H2A T120. While Survivin directly binds H3T3ph, H2A T120ph is recognized by Sgo1, which recruits Borealin. The manuscript by Abad et al discovered a novel role of Borealin on directly recognizing the nucleosome independently of mitotic histone phosphorylation. By reconstituting the minimum localization module (LM) of the CPC (Borealin, Survivin and INCENP 1- 58) using purified proteins in vitro, the authors demonstrated that the LM can associate with nucleosome core particles (NPCs) independently of H3T3ph or Sgo1, while H3T3ph increased its affinity. Through conducting systematic mutation analysis, the authors found that N-terminal 10 aa region, the middle loop domain, and the C-terminal dimerization domain all contributed to NCP-binding. Crosslinking MS analysis further suggested that the N-terminus is likely to interact with H2A/H2B acidic patch, while the middle loop domain may interact more broadly perhaps with core nucleosomal DNA. Localization analysis of Borealin mutants supports the importance of these nucleosome-interacting modules of Borealin for CPC targeting to mitotic chromosomes.

Overall, in vitro biochemical data with EMSA and SPR assays are compelling to support the authors' major conclusion that the CPC can directly interact with the nucleosome without mitotic phosphorylation through Borealin. However, cellular analyses need additional supporting data to match the JCB standard. In addition, descriptions for the crosslinking MS analysis are not sufficient to validate authors' interpretations. Despite these shortcomings, the manuscript will have significant impact toward understanding the mechanistic basis for the CPC regulation, after appropriate revisions.

Major points

1. Localization analysis of the CPC with Borealin mutants was solely based on transient transfection. Since a conflicting observation was reported by the Lens lab, showing (based on transient transfection) that fusing Survivin to INCENP bypassed the requirement for Borealin on centromere localization (PMID: 16239925), it would be important to establish the importance of Borealin in CPC localization at centromeres/chromosomes with a better experimental set-up.

1-1) Interpretations of the phenotypes caused upon transient transfection may not be straightforward without cautiously executed supporting evidence. Expression of expected products, relative expression level, and expected complex formation must be confirmed. In my personal view, most cell biologists in the mitosis field have stopped relying on transient transfection for localization analysis, because it is difficult to execute quantitative analysis, and thus to assure reproducibility. Since the major conclusion of this manuscript is that Borealin directly mediates nucleosome

interaction of the CPC, I feel that better cautious analysis on the localization analysis is necessary. The authors may consider generating inducible cell lines, for example using HeLa T-Rex.

1-2) Information about Myc-tagging is lacking. I assume that tandem copy of Myc has been added to N-terminus. Since the very N-terminus of Borealin contributes to nucleosome binding, it is important to describe the nature of the tag.

1-3) The quantitative analysis of immunofluorescence data is limited to Borealin 10-end (Fig.2D). Definition of "normalized Survivin intensity at centromeres" is not clearly explained.

2. Using crosslinking MS analysis, the authors conclude that Borealin employs multiple mechanisms to interact with NCP. However, the presented data are not sufficient to validate the authors' conclusion: the authors must show supporting data demonstrating that presented crosslinked residues represent physiological binding rather than surface accessibility. Several approaches would clarify this issue.

2-1) The authors must fully disclose peptides representing both intra- and inter-molecular crosslinks, including inter-molecular crosslinking within the LM. In addition, crosslinking analysis of LC alone and NCP alone must be conducted to validate the crosslinking mapping reflects known structures. This can then be compared with the crosslinking patterns of the LC-NCP complex. Evidence that the crosslinked LC-NCP complexes are not aggregated must be described.

2-2) The concentration (not just amount) of the protein complexes and buffer condition during crosslinking must be disclosed.

2-3) While C-terminal dimerization domain of Borealin contributes to nucleosome binding and centromere recruitment, it is not clear if this is through enhancing avidity, or directly interacting with the nucleosome. If the LC forms the dimer, it is possible that the dimeric LC can crosslinks NPCs, leading to aggregation, which makes the interpretation of crosslinking MS difficult. If the dimeric LC does not crosslink two NPCs, it would be also an important finding. The gel filtration pattern (Fig. S4A) unfortunately is not helpful.

2-4) The authors suggest that N-terminal 10 aa of Borealin interacts with the acidic patch of H2A/H2B. Since Borealin N-terminus is positively charged, this is a plausible hypothesis. Unfortunately, interpretation of Fig. 3B is confusing, as the Borealin residues shown as crosslinked to histone residues near acidic patch also belong to the middle region (K135, K158, K183, K198) and the C-terminal dimerization domain (K225, D245, E257, K263). Same Borealin residues are also shown to crosslink to other histone residues. For example, K135 and K158 can be crosslinked to three and four distinct histone residues, respectively. While some N-terminal residues in Borealin appear to be crosslinked to histone residues (Fig. 3A), details of these crosslinking patterns cannot be assessed with provided information. In short, it is impossible for me to evaluate the authors' conclusion that Borealin N-terminus contacts with acidic patch.

Minor points

1. To demonstrate that Survivin alone does not bind to NCP with H3T3ph, the level of phosphorylation must be shown. Western blotting and phos-tag gel analyses should be able to assess this point.

2. Klein et al (2006) have shown that Borealin can directly interact with DNA. This should be cited and discussed, related to the last sentence of the first paragraph in page 5.

3. Figure 2A. Each panel has two experimental segments, both noted as "CPC_LM". Is there any difference between two experimental conditions? Does the lane/column without no annotation indicates the wild-type LM? What does "Survivin" represent? Similar issues are also seen in Fig. S2B.

4. Fig. 4C. Some characters are too small to read.

5. Fig. S2A. Please define "CPC fl SPM".

6. Please revise the method section so that experiments can be readily reproduced. How was BL21(DE) pLysS cell line with three expression plasmids isolated? How was protein induction performed? It seems odd that both Survivin and Borealin were tagged with His, and the complex was purified with HisTrap. Is there any reason behind this strategy? Details of nucleosome formation procedures must be described. Primer sequence information for 601 amplification must be disclosed. What kind of method was used to evaluate formation of the mono-nucleosome? Transfection procedures must be described in detail including the amount of siRNA and expression plasmids.

Reviewer #3 (Comments to the Authors (Required)):

In this paper, the authors investigate how Borealin participates in recruiting the Chromosomal Passenger Complex (CPC) to chromosomes. The authors reconstitute CPC complexes containing Borealin, Survivin and truncated INCENP, and test the ability of various Borealin mutants to bind nucleosome core particles (NCPs). Up until now, binding of the CPC to centromeric nucleosomes has been suggested to be mediated through a direct interaction between pH3T3 and the BIR domain of Survivin and an incompletely-resolved pH2A-Sgo-Borealin interaction. Here, the authors demonstrate that both the N- and C-termini of Borealin are required for high affinity binding to nucleosomes in vitro and for CPC localization to centromeres in cells. They go on to show that Survivin is not sufficient to bind to NCPs in vitro, and that CPC complexes containing a Survivin BIR domain mutant retain the ability to bind NCPs. Based on their findings, the authors propose a new model for CPC recruitment to chromatin whereby direct binding between Borealin and the NCP constitutes the primary interaction mechanism. These findings should be of interest to the field, however, there are a few issues that should be addressed before consideration for publication in the JCB.

(1) The authors here demonstrate that both the N- and C-termini of Borealin are critical for CPC-NCP binding in vitro and for CPC recruitment to centromeres in cells. They also show that CPCs containing a mutant version of Survivin deficient for binding the phosphorylated H3 tail (Sur MUT) binds to NCPs as efficiently as wild-type complexes by EMSA, and they saw only a modest reduction (compared to the Borealin mutants) by SPR analysis. This suggests that Survivin's interaction with pH3T3 is not explicitly required for CPC-nucleosome binding. If this is the case, and direct interaction between Borealin and NCPs represents the major binding activity, is the Survivin BIR domain required for CPC recruitment to centromeres in cells? If the authors express the Survivin BIR-domain mutant in cells, do they see a much milder loss-of-CPC localization phenotype compared to their Borealin mutants? Given their new model for CPC localization, this should be explicitly tested in their assay.

(2) Are the authors trying to make the point that the CPC is excluded from both arms and centromeres in cells expressing Borealin mutants? This is not clear, since the authors interchange "chromatin," "chromosome," and "centromere" in the text and figure legends.

(3) The authors suggest a model in which "chromatin association of the CPC activates Haspin and

Bub1, which in turn phosphorylates H3 Thr3 and H2A Thr120, respectively." It is clear from previous studies that Aurora B phosphorylates Haspin to promote its activity and engages a feedback loop in which Haspin phosphorylation of pH3T3 promotes further association of the CPC with the centromere. Is this what the authors are referring to or some other mode of "activation"? It would help to clarify this point in the text.

(4) As the authors mention, Bub1 phosphorylation of histone H2A-T120 is implicated in Sgo1 recruitment, which is proposed to bind the CPC through an interaction with Borealin. The authors suggest here a second mode of Borealin binding to NCPs through a direct interaction with the acidic patch between H2A and H2B. It would be informative to determine how binding of the pH2A-Sgo1-recruited Borealin to NCPs impacts direct binding of Borealin to NCPs (through the newly identified contacts). I would expect that these binding experiments are outside of the scope of the current study, however, some discussion of how these two populations of Borealin might co-exist in the CPC-nucleosome complex would be helpful.

(5) The authors use surface plasmon resonance assays to demonstrate direct interactions between the CPC and nucleosome core particles. They describe in the methods fitting the data with a 1:1 stoichiometry. If Borealin is binding the acidic patch does it only have access to one of the two acidic patches on the nucleosome? And how does the dimerization domain of Borealin contribute to the stoichiometry?

(6) Why did the authors split the data for phospho-NCP and non-phospho-NCP binding assays between Figure 1 and Supp2? It is important to be able to compare the two conditions in same figure.

Reviewer #1 (Comments to the Authors (Required)):

Recruitment of the chromosome passenger complex (CPC), comprised of Borealin, Survivin, INCENP and Aurora B, to the inner centromere is required for proper chromosome segregation. The recruitment is thought to depend on a direct interaction of the CPC with the chromatin of the inner centromere. Understanding the biochemical and structural basis of CPC association with the nucleosome is important, as CPC recruitment to nucleosome is a critical mitotic event. In this short report, Abad et al. use biophysical and cell biological methods to address the mechanism for the chromosome association of CPC. In contrast to what was previously thought, the authors find that CPC binds chromatin in a histone modification independent manner, mainly via Borealin. Binding to H3T3ph is enhanced through Survivin. Based on MS-XL experiments, the authors propose that the interaction of the CPC with the nucleosome involves multiple contacts, especially between Borealin and H3. Based on these data the authors present a potential structural model of CPC recruitment to the nucleosome, although this model is not directly tested. The authors go on to suggest that instead of H3T3ph being upstream of CPC recruitment, that Borealin mediates nucleosome association of the CPC and activates Haspin and Bub1, which in turn phosphorylate H3 Thr3 and H2A Thr120, respectively. Subsequently, these histone phosphorylation marks increase the affinity of CPC for chromatin. The major strength of the manuscript is the strong and well executed biophysical undertaken to answer an important question and that it provides novel evidence for a more refined role for histone H3 modifications in CPC recruitment.

However, while the individual observations in the paper are very interesting, the study does not provide a compelling and integrated mechanism for how CPC binding to H3 nucleosome contributes to H3T3ph, and whether this process helps to restrict CPC recruitment.

We thank the reviewer for the positive evaluation of our work and for the constructive suggestions.

1. The cross-linking MS is a potentially very interested experiment; however, the manuscript does not provide concrete validation of the observations. The data are consistent with multiple interactions mediating the interaction between the CPC and the nucleosome. And the authors identify potential interaction regions in the NCP and the CPC. However, they do not explain the requirement for the loop region. Moreover, how meaningful the observations are is not clear because there is no in vitro or in vivo experimental validation.

We thank the reviewer for their suggestion to validate the intermolecular contacts we report between Borealin and nucleosomes (NCPs) based on the crosslinking MS (CLMS) experiments. We have now prepared several new mutant forms of CPC containing the following Borealin mutants and tested them in SPR experiments and in vivo siRNA-rescue assays: i) Borealin K12/R17/K20E – the residues contacting the acidic patch on nucleosomes, ii) Borealin K198/T199E and K135/K158/K183/K198/T199E, residues in the loop which likely directly contact DNA (as discussed below) and nearby histone residues iii) Borealin Δ 110-188 and Borealin Δ 189-206, the new loop deletions to identify the minimal loop region important for nucleosome binding.

Borealin K12/R17/K20E mutant reduced the NCP-binding ability of the CPC by about 2-fold ($K_d = 528 \pm 44.2$ nM vs 295.2 ± 40.9 nM) and the corresponding value for the H3T3ph NCP is about 5-fold ($K_d = 672.3 \pm 70.5$ nM vs 102.8 ± 34.2) and reduced the chromosomal association of the CPC in cells. We conclude that Borealin N-terminus interaction with the NCP acidic patch is critical for efficient NCP binding in vitro and chromosome association in vivo. We have now included this data in Fig. 4 and Fig. S3 and in the text on page 6.

As far as the requirement of the Borealin loop is concerned, while Borealin K198/T199E did not show significant reduction in NCP binding affinity (K_d 400 ± 30.3 nM), the Borealin K135/K158/K183/K198/T199E mutant showed a strong reduction, about 3-fold for the unmodified NCP and about 6-fold reduction for the H3T3ph NCP ($K_d = 1.1 \pm 0.2$ μ M and 570 ± 62.5 nM, respectively). Consistent with the SPR data, the Borealin K135/K158/K183/K198/T199E mutant showed reduced chromosome association of the CPC in cells. These data suggest that the N-terminal half of the Borealin loop (residues 110-188) contributes more for NCP binding as compared with the C-terminal half (residues 189-206). In agreement with this, CPC containing Borealin Δ 110-188 (CPC_LM Δ 110-188) bound NCPs relatively more weakly (1.8 ± 0.2 μ m) as compared with CPC containing Borealin Δ 189-206 (CPC_LM Δ 189-206; 260 ± 37 nm). We conclude that Borealin residues 110-188 is the minimal region of the loop crucial for nucleosome binding and amino acids K135, K158 and K183, found within this region, are the hot-spots mediating the interaction with nucleosomes. We have now included this data in Fig. 4, A, B and C and Fig.S3 E and F and added a paragraph in page 7.

What is the molecular basis for the contribution of the Borealin loop? Considering that the Borealin loop region is about 96 aa long and is predicted to be disordered, it is possible that some of the crosslinks observed between this loop region and the histone residues represent transient or/and non-specific interaction. As most Borealin loop contacts are near the DNA, (the theoretical pI of the loop is 10.4) and as Borealin has previously been proposed to bind DNA (Klein et al., 2006), we hypothesised that the Borealin loop region is making direct contact with nucleosomal DNA. To test this, we revisited the SPR sensorgrams obtained for the DNA binding of CPC_LM and CPC_LM Δ loop. By normalising the DNA binding to the steady state response for 1 μ M CPC_LM and by analysing the molecular weight (MW)-corrected response, we estimated the limit for the K_d values for the DNA-binding of CPC_LM and the CPC_LM Δ loop. CPC_LM bound DNA with a K_d of ≥ 5 μ M, while the corresponding value for the CPC_LM Δ loop was ≥ 20 μ M, indicating that the loop region is responsible for the binding of CPC to the DNA. Furthermore, we also tested the contribution of different Borealin loop regions to bind DNA in the SPR assays using recombinant Sumo-Borealin₁₁₀₋₂₀₆ (spanning the entire loop), Sumo-Borealin₁₁₀₋₁₈₈ (N-terminal half of the loop) and Sumo-Borealin₁₈₉₋₂₀₆ (C-terminal half of the loop). Supporting our hypothesis, Sumo-Borealin₁₁₀₋₂₀₆ bound DNA with a K_d of ≥ 5 μ M. However, removing N-terminal half of the loop, Borealin₁₈₉₋₂₀₆ bound DNA with a strongly reduced affinity ($k_d \geq 20$ μ M), while removing the C-terminal half of the loop, Sumo-Borealin₁₁₀₋₁₈₈, showed only a modest reduction in affinity ($k_d \geq 10$ μ M). Consistent with this the Borealin mutant harbouring point mutations within the C-terminal half of the Borealin loop (Borealin K198/T199E) did not show a noticeable reduction in the nucleosome binding ability of the CPC. However, when these mutations were combined with additional mutations within the N-terminal half of the loop (K135/K158/K183/K198/T199E) strongly reduced the nucleosome binding ability of the CPC as discussed above. We conclude that the

Borealin loop can directly bind DNA with interactions mainly involving the N-terminal half of the loop region (aa residues 110-188). This data has now been included in Fig. 4 D and E and in text on page 6 and 7.

2. The authors compare the in vitro binding of CPC_LM to the modified and unmodified H3 tail but based on the crosslinking MS there are many interactions, and the H3 tail may be inconsequential. H3 tail-less mutants should be tested to determine if H3 tail binding is required.

We thank the reviewer for their suggestion. We have now evaluated the requirement of H3 tail for CPC binding in EMSAs and SEC using NCPs reconstituted with H3 lacking the first 31 aa (NCP_{H3 Tail-less}). The CPC formed a robust complex with NCP_{H3 Tail-less} in SEC and EMSAs. This strengthens our conclusion that H3 tail is not essential for the CPC-NCP binding per se, but H3 tail interaction with Survivin is essential for increasing the affinity for H3T3ph NCPs and hence for the enrichment of CPC at centromeres. This data is now included in Fig. S1 D and E and in text on page 4.

3. The data in figure 4 suggest that CPC recruitment is required for H3T3ph and H2AT120ph. However, the current manuscript does not provide any mechanistic basis for this pathway. The Borealin loop mutant in figure 4 does not provide significant mechanistic insight into the feed forward mechanism, as it does not get recruited to the inner centromere irrespective of the T3 phosphorylation status.

We apologise that the mechanistic basis proposed in our manuscript was not clearly articulated. As the exclusion of CPC from chromosomes seen for the Borealin loop mutant (myc-Borealin_{Δloop}) results in a clear reduction in the H3T3ph level, we propose that the H3 T3ph-independent chromosome association of the CPC is likely a prior requirement for effective Haspin activation and hence the accumulation of the H3T3ph mark. This modification, possibly together with the H2AT120ph/hSgo1 connection, concentrates CPC at centromeres due its increased affinity for centromeric chromatin (enriched with H3T3ph and H2AT120ph marks). While we agree with the reviewer that additional experimental evaluations are likely to provide key mechanistic insights into the feed forward mechanism, we feel such a study is beyond the scope of this manuscript, which mainly reports the essential direct nucleosome-binding role of Borealin in facilitating the chromosome association of the CPC upstream of the histone modification(s) driven centromere enrichment.

4. EMSA and SPS results are somewhat inconsistent. BOR (10-end) and (10-221) mutants have no significant effect in nucleosome binding through EMSA, while they appear to be critical in SPR. Granted that the SPR is a much more quantitative measure, but the differences are very stark. In contrast, the 1-221 mutant shows an enhanced binding and is not commented on in the text. The mutant with the largest effect consistent across EMSA and SPR is the delta loop; however, there is not a clear mechanistic reason given for the importance of this domain.

We agree with the reviewer that there are some disagreements between the SPR and the EMSA analyses, in terms of the affinities (either implied or calculated). However, given the different solution conditions required for each, this orthogonal method discrepancy for some of the interactions studied is not unexpected. The fundamentals of the EMSA assay precludes having any ionic additions present during the run. If the interaction has a prevalence of ionic/salt-bridge driven contacts at the molecular interface, these will be somewhat strengthened, by the resulting loss of the hydrophobic effect, under these conditions. Conversely, those not facilitated by this type of molecular recognition will be somewhat weakened. In addition, the EMSA is more difficult to routinely use to generate quantitative values for affinities. The solution conditions used during the SPR experiments, do not suffer from the same restrictions and the ionic contribution to both the specific protein:nucleosome interaction and the minimisation of any non-specific interaction with the matrix and immobilised ligands should be minimised and less prevalent with this technique. This is likely to be the major contributor to the odd disparity between the two techniques. However, we would like to highlight that there is broad agreement between the SPR analysis and the other biochemical studies reported, and this further supports the view that the SPR sensor surfaces, and the interaction of ligands with the protein(s), are generating internally consistent values for the affinities, and are a good representative of the appropriate physiological interaction, that is also broadly supported by the cross-linking and the in vivo data. We have now included a sentence explaining this in the text on page 4.

We also thank this reviewer for raising the concern that Borealin 1-221 mutant (CPC_LM_{BOR 1-221}) appears to bind NCPs with an increased efficiency as compared with the CPC containing full length Borealin (CPC_LM) in EMSAs. In addition to the potential limitations of EMSA discussed above, we speculate that removing just the C-terminal region of Borealin perhaps exposes regions of CPC, which are otherwise inaccessible, facilitating additional non-specific interactions with NCPs or free DNA. We now explicitly mention this on page 4 and 5.

5. It is not clear whether the effects of CPC loss are on Haspin and/or Bub1 recruitment or on the activity of the enzymes. Bub1 and Haspin recruitment should be assayed. Bub1 dependency on H3Th3 phosphorylation should also be tested in the same manner as Haspin is tested for H2ATh120 phosphorylation in figure S5.

We agree with the reviewer that our data do not explain if the reduced H3T3ph and H2AT120ph observed with the Borealin mutant not capable of facilitating chromosomal association of the CPC is due to defective recruitment or reduced activity of Haspin and Bub1. While such an analysis is crucial for detailed mechanistic understanding of the pathway, as noted in our response to point # 3, we think it is beyond the scope of this manuscript – which primarily focuses on the mechanistic insights of the chromosomal association of the CPC and demonstrates the essential requirement of the Borealin-mediated and phosphorylation-independent chromosomal association of the CPC upstream of the histone modifications-dependent centromere enrichment. However, to address the reviewers concern regarding the potential Bub1 dependency on H3Thr3ph we have now performed Bub1 siRNA and checked for H3T3 levels. Our data shows that Bub1 siRNA does not affect H3T3ph levels confirming that H3Thr3ph is upstream of H2AT120 phosphorylation. We have included this data in Fig. S3 I and on page 8.

Minor issues:

6. The labeling of figure 2A makes it very confusing. It is not immediately clear that the Survivin lane lacks the Borealin and INCENP fragment. In addition, it would be helpful to have a schematic of these mutants.

We have now changed the labelling of the different mutants in Fig. 2 A and added a panel of schematic diagrams for the mutants used in the assays (shown in supplementary Fig. S2 B).

7. It would be useful to label to the N and C termini in the small diagrams of the CPC above the figure to better orient the reader.

We thank the reviewer for their suggestion. We tried to include this information in the small diagrams, but unfortunately including this would make the small diagrams more crowded and not legible. Moreover, as the triple helical bundle is formed with the N-terminal regions of Borealin and INCENP and the C-terminal region of Survivin, labelling would require both N and C on both ends leading to confusion. However, to address this valid concern we are referring to Fig. S1 C in the Figure legends where we show how the CPC subunits and their structural domains are linked.

8. Likewise, the insets in the model in figure 4 are very small and extremely difficult to read.

We have now enlarged the insets to make them more legible.

9. The pattern of ACA staining looks different in the Myc-Borealin 10-221 condition in Figure 2C suggesting the phenotype is distinct. The analysis of phenotypes of the different Borealin mutants is limited and could be expanded to include cell cycle, chromosome segregation or mitotic defects that might help distinguish to roles of the domains of Borealin.

We thank the reviewer for this suggestion. We have now quantified chromosome congression errors for all the Borealin mutants tested in Fig 2C. While all Borealin mutants that show reduced nucleosome binding result in an increase in the number of cells with uncongressed chromosomes, Myc-Borealin₁₀₋₂₂₁ showed a noticeably stronger phenotype with almost all cells showing more than one uncongressed chromosome. This suggests that the C-terminal region of Borealin, in addition to its contribution to NCP binding might have an additional role in ensuring proper chromosome congression. We have included the quantification data in Fig. 2 E and in the text on page 5.

10. The delta loop and 10-221 mutants should also be included in statistical part of figure 2C.

We have evaluated the centromeric levels of Survivin for both the CPC_LM_{Δloop} and the CPC_LM₁₀₋₂₂₁ and included them in Fig. 2 D.

Reviewer #2 (Comments to the Authors (Required)):

The Chromosomal Passenger Complex composed of Aurora B, Survivin, Borealin and INCENP plays multiple essential roles during mitosis. One of its key functions is to regulate kinetochore-microtubule attachment and centromeric cohesion through localizing at the centromere. It has been established that this centromeric enrichment of the CPC is mediated by mitosis-specific phosphorylation of two histone residues, H3T3 and H2A T120. While Survivin directly binds H3T3ph, H2A T120ph is recognized by Sgo1, which recruits Borealin. The manuscript by Abad et al discovered a novel role of Borealin on directly recognizing the nucleosome independently of mitotic histone phosphorylation. By reconstituting the minimum localization module (LM) of the CPC (Borealin, Survivin and INCENP 1- 58) using purified proteins in vitro, the authors demonstrated that the LM can associate with nucleosome core particles (NPCs) independently of H3T3ph or Sgo1, while H3T3ph increased its affinity. Through conducting systematic mutation analysis, the authors found that N-terminal 10 aa region, the middle loop domain, and the C-terminal dimerization domain all contributed to NCP-binding. Crosslinking MS analysis further suggested that the N-terminus is likely to interact with H2A/H2B acidic patch, while the middle loop domain may interact more broadly perhaps with core nucleosomal DNA. Localization analysis of Borealin mutants supports the importance of these nucleosome-interacting modules of Borealin for CPC targeting to mitotic chromosomes.

Overall, in vitro biochemical data with EMSA and SPR assays are compelling to support the authors' major conclusion that the CPC can directly interact with the nucleosome without mitotic phosphorylation through Borealin. However, cellular analyses need additional supporting data to match the JCB standard. In addition, descriptions for the crosslinking MS analysis are not sufficient to validate authors' interpretations. Despite these shortcomings, the manuscript will have significant impact toward understanding the mechanistic basis for the CPC regulation, after appropriate revisions.

We thank the reviewer for the positive evaluation of our work and for the constructive suggestions.

Major points

1. Localization analysis of the CPC with Borealin mutants was solely based on transient transfection. Since a conflicting observation was reported by the Lens lab, showing (based on transient transfection) that fusing Survivin to INCENP bypassed the requirement for Borealin on centromere localization (PMID: 16239925), it would be important to establish the importance of Borealin in CPC localization at centromeres/chromosomes with a better experimental set-up.

1-1) Interpretations of the phenotypes caused upon transient transfection may not be straightforward without cautiously executed supporting evidence. Expression of expected products, relative expression level, and expected complex formation must be confirmed. In my personal view, most cell biologists in the mitosis field have stopped relying on transient transfection for localization analysis, because it is difficult to execute quantitative analysis, and thus to assure reproducibility. Since the major conclusion of this manuscript is that Borealin directly mediates nucleosome interaction of the CPC, I feel that better cautious

analysis on the localization analysis is necessary. The authors may consider generating inducible cell lines, for example using HeLa T-Rex.

We thank the reviewer for raising this valid concern. We have now probed the expression levels and complex formation of the Borealin mutants using western blots and immunoprecipitation experiments, respectively. We confirm that all Borealin mutants are expressed and present at a similar level. Consistent with the in vitro reconstitutions, Borealin mutants showed complex formation with Survivin, INCENP and Aurora B in cells as assessed by the IPs. We have included this data in Fig. S2 E and F and a statement in page 5.

We agree with the reviewer that transient expression of mutant proteins may not be the best approach for localisation analysis. However, in this case we strongly believe that mislocalisation of the CPC observed with the various nucleosome-binding deficient Borealin mutants is not due to the over expression for the following reasons:

- The over expressed Borealin mutants not capable of binding nucleosome in vitro are completely excluded from chromosomes (together with other CPC components) suggesting that nucleosome binding affinities of these mutants are so low in the cells that even an increased abundance of Borealin mutants due to over expression is not sufficient to facilitate their chromosome association (Fig. 2 C).*
- Moreover, we do not see any aggregation or accumulation of CPC at any specific cellular site and these over expressed mutants are rather uniformly diffused throughout the cell (Fig. 2 C).*
- Most importantly, there is a clear correlation between the in vitro nucleosome binding and in vivo rescue assays - Borealin mutants not capable of binding nucleosome in vitro are excluded from chromosomes in cells.*
- In response to reviewer no 3, we have also evaluated the contribution of Survivin BIR – Histone H3 tail interaction for the chromosome association of the CPC in cells using a Survivin BIR mutant (K62/E65/H80A) not capable of binding the phosphorylated Histone H3 tail (Fig S1 G and H). In consistent with previous studies (Niedzialkowska E, et al., Mol Biol Cell 2012; Cao et al., Biochem Biophys Res Commun. 2006), our data clearly shows that this Survivin mutant does not abolish the chromosome association of the CPC and strengthens our major conclusion that Borealin-mediated nucleosome interaction is essential for the chromosome association of the CPC. However, to address the reviewer's concern regarding the over expression, we tried hard to generate stable lines for some of the Borealin mutants using HeLa T-Rex system. Unfortunately, we were not successful in generating the stable cell lines within the timeframe for revision due to technical difficulties: issues related to the integration of Borealin variants and the viability of the batch of HeLa TREX cells we obtained from collaborators. However, in the absence of stable cell line, we assessed if there is any correlation between the levels of Borealin Δ oop mutant expression and its inability to associate with chromosomes. As can be seen in the figure (Fig. R1, prepared for the purpose of this rebuttal) below, irrespective of the expression levels of the Borealin Δ oop mutant Borealin and other CPC subunits (Survivin) are excluded from chromosomes. We have now explicitly included a statement reflecting this on page 5.*

Figure R1. Representative images of cells expressing different levels of Myc-Borealin Δ loop in Borealin siRNA-depleted cells.

We also note that Vader et al., (PMID: 16239925), which claims that Survivin-INCENP chimera can rescue Borealin depletion and the role of Borealin is merely stabilising the Survivin-INCENP interaction, show noticeable level of residual Borealin after siRNA treatment (EMBO Rep, Vol 7, No 1, 2006. Fig 3F) and also states that ‘the association of the Survivin-INCENP chimera to centromeres and the central spindle was affected in Borealin-depleted cells’ (Page 88, first paragraph and first line).

1-2) Information about Myc-tagging is lacking. I assume that tandem copy of Myc has been added to N-terminus. Since the very N-terminus of Borealin contributes to nucleosome binding, it is important to describe the nature of the tag.

We have now included the details about the nature of the tag in the ‘Rescue experiments and Immunofluorescence Microscopy’ section of the materials and methods: ‘The pCDNA3.1 vector containing N-terminally tagged 3xMyc-Borealin was a gift from E. Nigg’s lab’.

1-3) The quantitative analysis of immunofluorescence data is limited to Borealin 10-end (Fig.2D). Definition of "normalized Survivin intensity at centromeres" is not clearly explained.

We have now included the quantification of the Borealin Δ loop and the Borealin₁₀₋₂₂₁ rescue experiments in Fig. 2 D. We have also elaborated the definition of ‘Normalised Survivin intensity at centromeres’ in the revised ‘Materials and methods’ section.

2. Using crosslinking MS analysis, the authors conclude that Borealin employs multiple mechanisms to interact with NCP. However, the presented data are not sufficient to validate the authors' conclusion: the authors must show supporting data demonstrating

that presented crosslinked residues represent physiological binding rather than surface accessibility. Several approaches would clarify this issue.

We thank the reviewer for raising this concern. As discussed in our response to Reviewer 1-point 1-1, we have now prepared several new Borealin mutants designed based on the CLMS data, and we show that Borealin K12/R17/K20E – the residues contacting the acidic patch on nucleosomes and Borealin K135/K158/K183/K198/T199E, which we believe directly contacts DNA based on the SPR studies, showed weakened affinity for NCPs and reduced chromosome association of the CPC. This new data is included in Fig 4 and discussed on pages 6 and 7. We think that this data together with those discussed below show that the specific crosslinks analysed in this study represent the physiological binding of CPC with nucleosomes.

2-1) The authors must fully disclose peptides representing both intra- and inter-molecular crosslinks, including inter-molecular crosslinking within the LM. In addition, crosslinking analysis of LC alone and NCP alone must be conducted to validate the crosslinking mapping reflects known structures. This can then be compared with the crosslinking patterns of the LC-NCP complex. Evidence that the crosslinked LC-NCP complexes are not aggregated must be described.

All datasets generated in this study are deposited in the PRIDE database. To address the above point, we have now performed new CLMS experiment for CPC_LM and NCP on their own. As seen below in Fig. R2 prepared for the purpose of the rebuttal, overall pattern of crosslinks observed between the subunits of the CPC_LM and NCP on their own and in the context of CPC-NCP complex is very similar. We speculate that any difference might represent conformational variations associated with the complex formation. We also note that the inter-subunit crosslinks observed generally agree with the available three-dimensional structures. We have mentioned this in the text on page 5, but not included the figure due to space constraint.

Figure R2. (A) Circular view of crosslinks observed between the CPC subunits in the context of (a) CPC (b) CPC-NCP complex and the core histones in the context of (c) NCP and (d) CPC-NCP complex. Crosslinks mapped onto the structures of (B) the CPC core, (C) the Borealin dimerization domain and (D and E) Nucleosome Core Particle. The lines represent contacts derived using a C α -C α distance constraint of 20Å (based on Merkley et al., *Protein Sci* 2014, 23(6), 747-59 and Yoon et al., *JMB* 2018, 430 (6), 822-841) (E) Crosslinks mapped onto the structure of Nucleosome Core Particle without C α -C α distance constraint to highlight the contacts between the flexible N-terminal Histone tails and the core histones.

Furthermore, we appreciate that in solution-crosslinking reactions can stabilise weak interparticle interaction leading to false positive contacts. To ensure that the crosslinked CPC-NCP complex is not aggregated, we clarified the crosslinking reaction by introducing a centrifugation step prior to SDS PAGE and MS analysis. The crosslinks observed with and without the centrifugation step were nearly identical and involve mostly the same regions (Please see Fig. R3). Due to space constraints we decided not to include the below Fig. R3 in the revised manuscripts.

Figure R3. Circular view of crosslinks observed for the CPC-NCP complex without (A) and with (B) an additional centrifugation step prior to the SDS-PAGE/MS analysis.

2-2) The concentration (not just amount) of the protein complexes and buffer condition during crosslinking must be disclosed.

We thank the reviewer for spotting this. We have now included this information in the 'Materials and methods' section.

2-3) While C-terminal dimerization domain of Borealin contributes to nucleosome binding and centromere recruitment, it is not clear if this is through enhancing avidity, or directly interacting with the nucleosome. If the LC forms the dimer, it is possible that the dimeric LC can crosslinks NPCs, leading to aggregation, which makes the interpretation of crosslinking MS difficult. If the dimeric LC does not crosslink two NPCs, it would be also an important finding. The gel filtration pattern (Fig. S4A) unfortunately is not helpful.

We agree with the reviewer that understanding the subunit stoichiometry of CPC-NCP binding is important to fully understand the structural basis. We tried extensively to measure the molecular weight (MW) of the CPC-NCP in the Size Exclusion Chromatography combined Multi-Angle Light Scattering (SEC-MALS) Experiment. Unfortunately, the CPC-NCP did not behave well (despite performing the experiments under different buffer conditions and

concentrations) to reliably estimate the MW and hence the stoichiometry of binding, possibly due to the temperature at which the MALS system is operated. However, we estimated the stoichiometry of CPC-NCP binding using steady state RU_{max} values observed in the SPR. For CPC_{LM} binding to NCP, the steady state RU_{max} values observed experimentally (~ 275 RU; Fig. 1 F) are very close to the theoretical RU_{max} values expected for a 1:1 stoichiometric interaction (approximately 250 – 300 RU) between a CPC_{LM} dimer (100 kDa) and an NCP (200 kDa), given the molecular weight ratio of the interacting components and the amount of nucleosome immobilised on the sensor surface (500 RU). Moreover to understand the contribution of the Borealin dimerization domain we tested NCP binding ability of the Borealin dimerisation domain (Sumo-Borealin₂₂₂₋₂₈₀) in SPR which did not show any binding. Based on this and the estimated stoichiometry of binding, we speculate that Borealin dimerization domain likely increases the affinity by modulating the interaction by providing two copies of CPC that can interact with the acidic patch and H3 tail present on either side of a single NCP. We have now included the data in Fig 4F and in the text on page 7 and 8.

2-4) The authors suggest that N-terminal 10 aa of Borealin interacts with the acidic patch of H2A/H2B. Since Borealin N-terminus is positively charged, this is a plausible hypothesis. Unfortunately, interpretation of Fig. 3B is confusing, as the Borealin residues shown as crosslinked to histone residues near acidic patch also belong to the middle region (K135, K158, K183, K198) and the C-terminal dimerization domain (K225, D245, E257, K263). Same Borealin residues are also shown to crosslink to other histone residues. For example, K135 and K158 can be crosslinked to three and four distinct histone residues, respectively. While some N-terminal residues in Borealin appear to be crosslinked to histone residues (Fig. 3A), details of these crosslinking patterns cannot be assessed with provided information. In short, it is impossible for me to evaluate the authors' conclusion that Borealin N-terminus contacts with acidic patch.

We agree with the reviewer the crosslinking MS analysis presented in the original manuscript has its limitation in explaining the precise level of contribution of multiple regions of Borealin for NCP binding. As mentioned in the response to point 2-1 above, we appreciate that in-solution crosslinking reactions can stabilise weak interparticle interactions (mostly involving the flexible unstructured regions) leading to false positive contacts. While we think cryoEM/crystal structure of CPC-NCP is needed to understand the high-resolution structural basis of CPC-NCP assembly, our additional biochemical and siRNA rescue experiments performed with the new CLMS-based Borealin mutants (as addressed in response to Reviewer 1, point 1-1 and Reviewer 2, point 2-1) allowed us to propose the mode of contribution of various Borealin regions (N-terminal region, Loop and the dimerization domain) for CPC-NCP binding: i) Borealin N-terminal region interacts with the NCP acidic patch ii) Borealin loop mainly contacts the DNA and possibly neighbouring histone residues and iii) Borealin dimerization enhances CPC-NCP affinity by facilitating the full occupation of the CPC-interaction sites (acidic patch, DNA and H3 tail) related by the intrinsic two fold symmetry of the NCP. We have discussed this explicitly in the text on page 8.

Minor points

1. To demonstrate that Survivin alone does not bind to NCP with H3T3ph, the level of

phosphorylation must be shown. Western blotting and phos-tag gel analyses should be able to assess this point.

H3T3ph NCPs are reconstituted using native chemical ligation as described in the 'Materials and methods' and contain homogeneous H3T3ph modification: 'Native chemical ligation reactions with H3Δ1–31 MT32C C110A and the N-terminal H3 peptide ARTPhKQTARKSTGGKAPRKQLATKAARKSAPA containing a C-terminal benzyl thioester (Peptide Protein Research Ltd., Fareham, UK) were carried out in 6 M Guanidine HCl, 250 mM sodium phosphate buffer pH 7.2, 150 mM 4-mercaptophenylacetic acid (MPAA), 50 mM TCEP for 72 h at room temperature with constant agitation. Reactions were dialysed three times against 7 M urea, 100 mM NaCl, 10 mM Tris pH 8, 1 mM EDTA, 1 mM DTT. Ligated full-length H3T3Ph histone was separated from unligated truncated histone through cation exchange chromatography on a monoS column (GE) and then dialysed against water containing 5 mM β-mercaptoethanol before lyophilization and storage at -80°C.'

2. Klein et al (2006) have shown that Borealin can directly interact with DNA. This should be cited and discussed, related to the last sentence of the first paragraph in page 5.

We thank the reviewer for this suggestion. We have now cited this work on page 6.

3. Figure 2A. Each panel has two experimental segments, both noted as "CPC_LM". Is there any difference between two experimental conditions? Does the lane/column without no annotation indicates the wild-type LM? What does "Survivin" represent? Similar issues are also seen in Fig. S2B.

As addressed to Reviewer 1, minor suggestion 6, we have now changed the format of labelling to resolve this.

4. Fig. 4C. Some characters are too small to read.

We have now enlarged the characters and the corresponding figure in the revised manuscript is Fig. 5 C.

5. Fig. S2A. Please define "CPC fl SPM".

We apologise for the inconsistency. 'CPC fl SPM' should have read as CPC_LM_{SUR MUT}. We have now amended this in Fig. S2 A.

6. Please revise the method section so that experiments can be readily reproduced. How was BL21(DE) pLysS cell line with three expression plasmids isolated? How was protein induction performed? It seems odd that both Survivin and Borealin were tagged with His, and the complex was purified with HisTrap. Is there any reason behind this strategy? Details of nucleosome formation procedures must be described. Primer sequence information for 601 amplification must be disclosed. What kind of method was used to evaluate formation of the mono-nucleosome? Transfection procedures must be described in detail including the amount of siRNA and expression plasmids.

We have now included the following information in the materials and methods: 'The CPC subunits were co-expressed in BL21(DE) pLysS strain by co-transforming the 3 vectors containing the individual CPC components. Cultures were induced overnight at 18°C'.

We confirm that we used both Survivin and Borealin as his-tagged for the reconstitution of the CPC complex. After several trials, we found that this combination, together with untagged INCENP 1-58 led to the purest protein with the highest yield.

We have also extended the information regarding the NCP reconstitution as follows: A pBS-601 Widom vector was used to amplify the 147bp 601 Widom positioning sequence with unlabeled, 5' IR700- or biotin-labelled primers (forward: 5'-ACA GGA TGT ATA TAT GTG ACA CG-3' and reverse: 5'-CTG GAG AAT CCC GGT GCC-3'). Mononucleosomes were obtained by using the salt gradient dialysis method. After optimization of the octamer-DNA ratio by analysing the nucleosomes using a 6 % acrylamide native gel, the histone octamer-DNA mix was dialysed against TE buffer (10 mM Tris pH 8, 1 mM EDTA, 50 mM NaCl) overnight by gradually decreasing the ionic strength from 2 M using a peristaltic pump.

We have also included the amount of siRNA and expression plasmids in the appropriate section of the 'Materials and Methods'.

Reviewer #3 (Comments to the Authors (Required)):

In this paper, the authors investigate how Borealin participates in recruiting the Chromosomal Passenger Complex (CPC) to chromosomes. The authors reconstitute CPC complexes containing Borealin, Survivin and truncated INCENP, and test the ability of various Borealin mutants to bind nucleosome core particles (NCPs). Up until now, binding of the CPC to centromeric nucleosomes has been suggested to be mediated through a direct interaction between pH3T3 and the BIR domain of Survivin and an incompletely-resolved pH2A-Sgo-Borealin interaction. Here, the authors demonstrate that both the N- and C-termini of Borealin are required for high affinity binding to nucleosomes in vitro and for CPC localization to centromeres in cells. They go on to show that Survivin is not sufficient to bind to NCPs in vitro, and that CPC complexes containing a Survivin BIR domain mutant retain the ability to bind NCPs. Based on their findings, the authors propose a new model for CPC recruitment to chromatin whereby direct binding between Borealin and the NCP constitutes the primary interaction mechanism. These findings should be of interest to the field, however, there are a few issues that should be addressed before consideration for publication in the JCB.

We thank the reviewer for the positive evaluation of our work and for the constructive suggestions.

(1) The authors here demonstrate that both the N- and C-termini of Borealin are critical for CPC-NCP binding in vitro and for CPC recruitment to centromeres in cells. They also show that CPCs containing a mutant version of Survivin deficient for binding the phosphorylated H3 tail (Sur MUT) binds to NCPs as efficiently as wild-type complexes by EMSA, and they saw only a modest reduction (compared to the Borealin mutants) by SPR analysis. This suggests that Survivin's interaction with pH3T3 is not explicitly required for CPC-nucleosome binding. If this is the case, and direct interaction between Borealin and NCPs represents the major binding activity, is the Survivin BIR domain required for CPC recruitment to centromeres in cells? If the authors express the Survivin BIR-domain mutant in cells, do they see a much milder loss-of-CPC localization phenotype compared to their Borealin mutants? Given their new model for CPC localization, this should be explicitly tested in their assay.

We thank the reviewer for raising this point. As suggested, we have now evaluated the contribution of the Survivin BIR domain interaction with H3 tail in siRNA rescue experiments. In consistent with previous studies (Niedzialkowska E, et al., Mol Biol Cell 2012; Cao et al., Biochem Biophys Res Commun. 2006), rescue of Survivin siRNA using a Survivin BIR mutant (K62/E65/H80A) not capable of binding the phosphorylated Histone H3 tail did not abolish the chromosome association of the CPC. This experiment strengthens our major conclusion that Borealin-mediated nucleosome interaction is essential for the chromosome association of the CPC. We have now included this data in Fig S1 G and H and in the text on page 4. We also quantified the chromosome congression defects for this mutant and the phenotype is relatively milder as compared with the phenotype observed for the Borealin mutant (Myc-Borealin Δ loop) which completely excludes CPC from chromosomes (included in Fig S1G).

(2) Are the authors trying to make the point that the CPC is excluded from both arms and centromeres in cells expressing Borealin mutants? This is not clear, since the authors interchange "chromatin," "chromosome," and "centromere" in the text and figure legends.

We apologise if our choice of words had affected the clarity of what we intended to convey. To answer the specific question of the reviewer: Yes, our data show that the Borealin mutants affecting the NCP binding of the CPC exclude CPC from chromosomes (both arms and centromeres). We have now gone through and edited the text carefully to avoid any possible confusion.

(3) The authors suggest a model in which "chromatin association of the CPC activates Haspin and Bub1, which in turn phosphorylate H3 Thr3 and H2A Thr120, respectively." It is clear from previous studies that Aurora B phosphorylates Haspin to promote its activity and engages a feedback loop in which Haspin phosphorylation of pH3T3 promotes further association of the CPC with the centromere. Is this what the authors are referring to or some other mode of "activation"? It would help to clarify this point in the text.

We agree, as noted by this reviewer, we refer to the Aurora B phosphorylation mediated enhancement of Haspin activity as the 'activation' of CPC. To make this clear we have rephrased this sentence as "Chromatin association of the CPC activates Haspin and Bub1 through Aurora B mediated Haspin phosphorylation and Bub1 recruitment (Krenn et al., front Oncol. 2015; 5:255 and Hindriksen et al., Front Cell Dev Biol., 2017), which in turn phosphorylate H3 Thr3 and H2A Thr120, respectively." on page 9.

(4) As the authors mention, Bub1 phosphorylation of histone H2A-T120 is implicated in Sgo1 recruitment, which is proposed to bind the CPC through an interaction with Borealin. The authors suggest here a second mode of Borealin binding to NCPs through a direct interaction with the acidic patch between H2A and H2B. It would be informative to determine how binding of the pH2A-Sgo1-recruited Borealin to NCPs impacts direct binding of Borealin to NCPs (through the newly identified contacts). I would expect that these binding experiments are outside of the scope of the current study, however, some discussion of how these two populations of Borealin might co-exist in the CPC-nucleosome complex would be helpful.

We thank the reviewer for their comment. As the reviewer had appreciated, probing how Borealin mediated nucleosome binding of CPC fits with/impacts pH2A-Sgo1-Borealin mediated CPC is beyond the scope of this manuscript. In the absence of any detailed molecular/structural insights into how Sgo1 interacts with or connects pH2A and Borealin, we do not feel comfortable making any speculations. However, as the reviewer had pointed out investigating this question in the future is important to fully understand the co-existence of multiple pathways to control CPC localisation and function. We have explicitly mentioned this in the revised manuscript on page 9.

(5) The authors use surface plasmon resonance assays to demonstrate direct interactions between the CPC and nucleosome core particles. They describe in the methods fitting the data with a 1:1 stoichiometry. If Borealin is binding the acidic patch does it only have access to one of the two acidic patches on the nucleosome? And how does the dimerization domain of Borealin contribute to the stoichiometry?

We thank the reviewer for raising these valid questions. Due to the complex nature of the interaction, and the, in some cases, clear multiphasic nature of the interaction between the

CPC constructs and the nucleosomes, kinetic models were not used in the fitting process. Almost all the interactions studied are well fit by a simple steady-state interaction model. In addition, the RU_{max} values determined are very close to the theoretical maximum response expected for a 1:2 (nucleosome:CPC) stoichiometric interaction given the molecular weight ratios of the interaction components and the amount of nucleosome immobilized (please see also our response to Reviewer 2 point 2-3). The χ^2 values for the fits are also good; most of them well below 2, resulting in excellent residuals and well within the accepted limits for “good fits”. One nucleosome binding per CPC dimer suggests Borealin dimerization facilitates the engagement of all ‘two-fold symmetry related’ CPC-interaction sites on the nucleosome. We have discussed this in the ‘Materials and methods’ and in the text on page 7 and 8.

(6) Why did the authors split the data for phospho-NCP and non-phospho-NCP binding assays between Figure 1 and Supp2? It is important to be able to compare the two conditions in same figure.

We thank the reviewer for the suggestion. We have now combined the EMSA analysis carried out for unmodified and H3T3ph NCPs in the same panel (Fig. 2 A).

September 6, 2019

RE: JCB Manuscript #201905040R

Dr. A. Arockia Jeyaprakash
University of Edinburgh
Wellcome Trust Centre for Cell Biology
Max Born Crescent
Edinburgh EH9 3BF
United Kingdom

Dear Dr. Jeyaprakash,

Thank you for submitting your revised manuscript entitled "Borealin-Nucleosome Interaction Secures Chromosome Association of the Chromosomal Passenger complex". Based on our editorial assessment of the revised manuscript and rebuttal, we find that the revision satisfactorily addressed the points of major concern. We have a few minor points for your consideration below as well as edits needed to meet our formatting requirements for publication. We would be happy to publish your paper in JCB pending final revisions necessary to address these points (see details below).

1) In Figure 4, D-F, the "greater than or equal to" values from the SPR analysis of affinity seem confusing. It is difficult to compare two "greater than or equal to" values because that doesn't make mathematical sense. Is there a different way to present this? (or potentially present normalized binding and indicate it is too weak to accurately measure affinities?) We are concerned readers will find this confusing too and this also needs to be addressed in the text. When specific affinities are derived from SPR analysis, this is not an issue.

2) page 5 line 5 from top - should be micromolar (μM) and not micrometer (μm). There may be other instances of this.

3) Arshad indicated, that, while reading the revision, one question that came to mind was whether the delta loop mutant of Borealin affected Ser10 (and possibly also Ser28) phosphorylation of histone H3. Inhibiting haspin does not affect Ser10 phosphorylation and the determinants needed for Ser10 phosphorylation are (at least to Arshad) unclear (aside from kinase activity of Aur B being critical). Was Ser10 phosphorylation checked in the Borealin delta loop mutant that delocalizes the CPC from chromosomes? Either outcome would be informative to mention. If Ser10 phosphorylation was reduced/inhibited, Arshad would highly recommend including that in the manuscript.

4) Figure formatting:

Molecular weight or nucleic acid size markers must be included on all gel electrophoresis. Please add molecular weight with unit labels on the following panels: S1E, S1H unit labels, S2AEF unit labels, S3ABEFG unit labels

5) Statistical analysis: Error bars on graphic representations of numerical data must be clearly described in the figure legend. The number of independent data points (n) represented in a graph must be indicated in the legend. Statistical methods should be explained in full in the materials and methods. For figures presenting pooled data the statistical measure should be defined in the figure legends.

Please indicate n/sample size/how many experiments the data are representative of: 2B, 4DEF

6) Materials and methods: Should be comprehensive and not simply reference a previous publication for details on how an experiment was performed. Please provide full descriptions in the text for readers who may not have access to referenced manuscripts.

- For instance, please provide more info as to how mononucleosomes were obtained, even if described in other published work.

- Microscope image acquisition: The following information must be provided about the acquisition and processing of images:

a. Make and model of microscope

b. Type, magnification, and numerical aperture of the objective lenses

c. Temperature

d. imaging medium

e. Fluorochromes

f. Camera make and model

g. Acquisition software

h. Any software used for image processing subsequent to data acquisition. Please include details and types of operations involved (e.g., type of deconvolution, 3D reconstitutions, surface or volume rendering, gamma adjustments, etc.).

7) References: There is no limit to the number of references cited in a manuscript. References should be cited parenthetically in the text by author and year of publication.

- Please abbreviate the names of journals according to PubMed.

- Please be sure to adequately format preprint references, per JCB formatting style:

<http://jcb.rupress.org/reference-guidelines>

(both in-text and in the reference list)

- Please format the refs per JCB style (no numbering)

8) A summary paragraph of all supplemental material should appear at the end of the Materials and methods section.

9) Conflict of interest statement: JCB requires inclusion of a statement in the acknowledgements regarding competing financial interests. If no competing financial interests exist, please include the following statement: "The authors declare no competing financial interests." If competing interests are declared, please follow your statement of these competing interests with the following statement: "The authors declare no further competing financial interests."

A. MANUSCRIPT ORGANIZATION AND FORMATTING:

Full guidelines are available on our Instructions for Authors page, <http://jcb.rupress.org/submission-guidelines#revised>. **Submission of a paper that does not conform to JCB guidelines will delay the acceptance of your manuscript.**

B. FINAL FILES:

-- High-resolution figure and video files: See our detailed guidelines for preparing your production-ready images, <http://jcb.rupress.org/fig-vid-guidelines>.

Thank you for this interesting contribution, we look forward to publishing your paper in the Journal of Cell Biology.

Sincerely,

Arshad Desai, PhD
Editor, Journal of Cell Biology

Melina Casadio, PhD
Senior Scientific Editor, Journal of Cell Biology